# The functional brain favours segregated modular connectivity at old age unless affected by neurodegeneration

Xue Chen [1,2,9 ✉], Joe Necus [2,3,9 ✉], Luis R. Peraza [4,5,9], Ramtin Mehraram[2,4,6,7,9], Yanjiang Wang[1], John T. O'Brien [8], Andrew Blamire [4], Marcus Kaiser[2,3,4] & John-Paul Taylor[4]

Brain's modular connectivity gives this organ resilience and adaptability. The ageing process alters the organised modularity of the brain and these changes are further accentuated by neurodegeneration, leading to disorganisation. To understand this further, we analysed modular variability—heterogeneity of modules—and modular dissociation—detachment from segregated connectivity—in two ageing cohorts and a mixed cohort of neurodegenerative diseases. Our results revealed that the brain follows a universal pattern of high modular variability in metacognitive brain regions: the association cortices. The brain in ageing moves towards a segregated modular structure despite presenting with increased modular heterogeneity—modules in older adults are not only segregated, but their shape and size are more variable than in young adults. In the presence of neurodegeneration, the brain maintains its segregated connectivity globally but not locally, and this is particularly visible in dementia with Lewy bodies and Parkinson's disease dementia; overall, the modular brain shows patterns of differentiated pathology.

[1] College of Control Science and Engineering, China University of Petroleum (East China), Qingdao, China. [2] Interdisciplinary Computing and Complex BioSystems (ICOS) research group, School of Computing, Newcastle University, Newcastle upon Tyne, United Kingdom. [3] University of Nottingham, NIHR Nottingham Biomedical Research Centre, School of Medicine, Nottingham, UK. [4] Translational and Clinical Research Institute, Newcastle University, Campus for Ageing and Vitality, Newcastle upon Tyne, United Kingdom. [5] IXICO Plc, London, UK. [6] Experimental Oto-rhino-laryngology (ExpORL) Research Group, Department of Neurosciences, KU Leuven, Leuven, Belgium. [7] NIHR Newcastle Biomedical Research Centre, Campus for Ageing and Vitality, Newcastle upon Tyne, UK. [8] Department of Psychiatry, University of Cambridge School of Medicine, Cambridge, United Kingdom. [9] These authors contributed equally: Xue Chen, Joe Necus, Luis R Peraza, Ramtin Mehraram. ✉email: wmtmdlove@163.com; Joseph.necus@nottingham.ac.uk

The brain can now be studied by a range of neuroimaging and electrophysiological technologies that allow us to further our understanding of its structure and function. By these recent developments, we know that brain function is dictated by complex interactions between neurons in the micro-, meso- and macroscales[1], and that these are shaped in a functional network with specific properties and characteristics[2,3]. The functional brain network is small-world, meaning that its structure reflects a balance between efficient communication and wiring cost[4,5]. The brain achieves this efficiency by creating groups of neurons densely connected among themselves but loosely communicated between the different groups; these are typically referred to as communities or modules[6]. It is hypothesised that the modularity of the brain is the result of evolution, where, in an always changing environment the brain developed a strategy to adapt subsystems rapidly without compromising the totality of its network[6]. In this regard, recent studies on brain dynamics have reported that modules constrain dynamic communication within their boundaries without affecting other modules[5,7,8]. This property also gives the brain superior resilience against attacks either by disease or injury[4,9].

Previous research in brain modularity has reported, consistently, several major modules such as the motor-sensory, visual and default-mode modules[10,11]. Although there is no agreement on their number, the majority of functional studies report that between three to ten modules are present in the brain[6,12]. This number depends on many factors and methodologic preferences such as the number of brain regions (brain parcellation), the chosen brain atlases[13], neuroimaging pre-processing pipelines[14], connectivity measures (e.g. wavelet or Pearson correlations), and treatment of connectivity weights[15]. Regardless of these differences, it is now agreed that brain modules are variable across time and between individuals, changing in shape and size depending on the cognitive task or no task at all, as is the case of the resting state. A recent investigation by Bassett, et al.[16] in a task-based functional magnetic resonance imaging (fMRI) study found that after training, brain modules become more segregated and this was linked to learning and specialisation of regions. In the same fashion, Baum, et al.[17] reported that the modular brain becomes segregated during development, from childhood to adulthood, a finding reported as well by others[18]. From the perspective of diseases, alterations to the brain's modularity depend highly on the disease. In schizophrenia, for instance, there is a decrease in modularity suggesting leakage of information between modules[12] while in Lewy body diseases there is an increase in modularity suggesting segregated communities[19,20]. Certainly, the processes of ageing and neurodegeneration can alter brain communities; however how this is done and how the brain diverges from healthy ageing to a dementia state due to neurodegeneration is still not completely understood.

To address this issue, we studied functional modular changes by the ageing process within two public databases; the Enhanced Nathan Kline Institute Rockland Sample (NKI, $N = 297$ participants)[21] and the 1000 Functional Connectomes Project (TFC, $N = 359$)[22]. Additionally, to test the effects of neurogeneration on brain's communities we also analysed a Newcastle University (NCL) database of neurodegenerative dementias[23–25] which comprised an Alzheimer's disease dementia group (ADD, $N = 42$), and two Lewy body disease groups which included both dementia with Lewy bodies (DLB, $N = 38$) and Parkinson's disease dementia (PDD, $N = 17$). These diseases are the most common cause of neurodegenerative dementia in older adults[26] and represent a spectrum between a more cortical amyloid disease and subcortical alpha-synuclein disease that often concur in patients[27]. The NCL database also included age-matched healthy participants ($N = 34$).

We investigated group modular variability (MV) which measures the heterogeneity of network communities across participants[28]. Additionally, we studied a proposed measure we have named modular dissociation (MD). MD is defined as the community difference between two networks constructed by global and local thresholding of network connections. The global threshold is the standard and most used method for network construction while local thresholding is an alternative approach that favours segregation of modules by connecting the $k$ nearest neighbours of a node, i.e. the strongest $k$ connections. Hence, MD measures how close a globally thresholded network is from the nearest neighbour connectivity regime that favours modular segregation; thus high MD values indicate that the networks dissociate from the nearest neighbour connectivity regime.

We compared differences in MV and MD between young and older adults from the NKI and TFC cohorts and the deviation from healthy ageing to disease with the NCL cohort. This analysis was performed at optimal edge density (network cost) using 451 region-of-interest (ROI) functional atlas[29]. For validation purposes, group MV and MD were also estimated at 10 and 20% edge densities, and with three additional functional atlases; 100, 200, 247 ROI. A schematic figure showing the estimation procedure is shown in Fig. 1.

Our results show that the brain has a consistent pattern of high MV and MD across all studied cohorts which involves the higher association cortices and basal brain structures respectively, indicating a topographic and preserved universality to these measures/patterns. These patterns also demonstrated a consistent change as a result of healthy ageing, as observed in both independent ageing cohorts, where the brain moves towards a more segregated modular network structure that favours nearest neighbour connectivity despite presenting with increased modular heterogeneity, with the exception of the insulo-opercular cortex which showed higher values of dissociation and heterogeneity in older adults. However, in the presence of neurodegenerative dementia, the brain's dissociation remains invariant—globally it stays in a segregated state, which is particularly pronounced in DLB and PDD. In contrast, it is heavily altered at the modular level with decreased MD in frontal-related modules and increased MD in the motor-sensory module. The present work represents an advancement in the understanding of the effects of ageing in the modular brain and the changes driven by neurodegenerative diseases on the ageing brain.

## Results

Participants from the ageing neuroimaging databases were selected and divided by age: young adults (YA) between 20–40 years old (NKI = 151 and TFC = 257 participants) and older adults (OA) between 60–80 years old (NKI = 146 and TFC = 102 participants). The healthy participants within the NCL neuroimaging database were classified as an independent OA group and used as a reference for comparisons with the three dementia groups. Demographics of all groups are given in Supplementary Table 1.

**Length-vs-strength connectivity behaviour is not different among dementias.** We examined edge distance vs edge strength (weight) behaviour within each group and performed comparisons between groups for each of the three cohorts in this study. Weights for this analysis belonged to the network connections that survived optimal edge density using local threshold network construction (Supplementary Fig 2). We implemented a linear model to test for differences in the intercept and slope between two groups with connectivity strength as the dependent variable. In both ageing cohorts (TFC and NKI) a difference in the

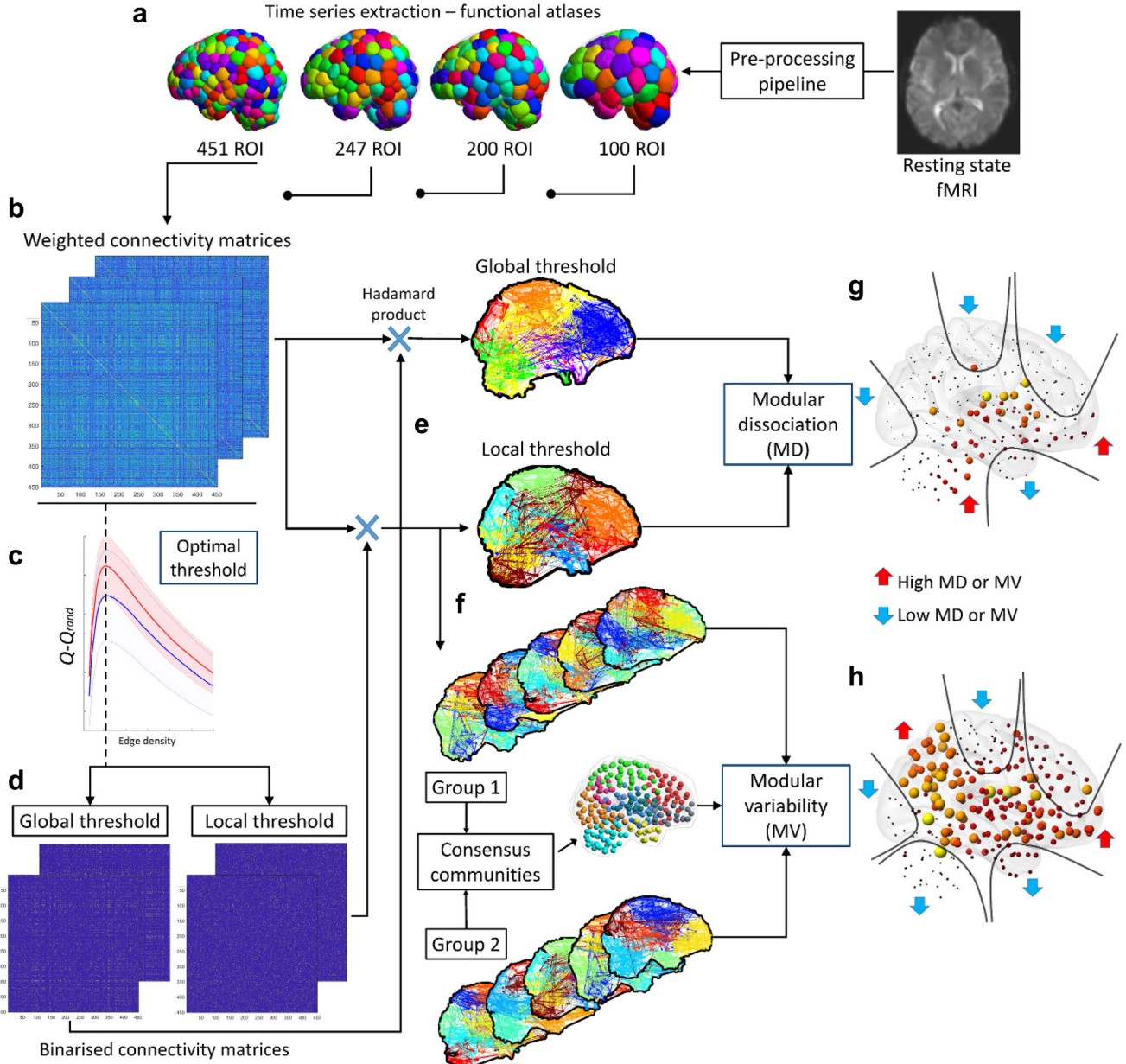

**Fig. 1 Methods for modular variability (MV) and modular dissociation (MD). a** Resting-state functional MRI pre-processing pipeline and time-series extraction from functional atlases. **b** Pearson correlation matrices. **c** Optimal local threshold estimation; the network edge density at which $Q-Q_{rand}$ is maximum. **d** Thresholded matrices by optimal density using local and global threshold network construction methods. **e** Louvain's community and modular dissociation (MD) estimation, see also Supplementary Fig 1. **f** Modular variability (MV) using consensus community. **g** Group means MD; subcortical regions and cerebellum showed in all groups high MD while motor-sensory, frontal, temporal pole and occipital cortex show low MD. **h** Group mean MV; patterns of high and low MV were consistent across all groups. Motor-sensory, occipital, and temporal pole showed low MV while parietal, ventral frontal and insulo-opercular cortices showed high MV.

strength-vs-distance slope was found. Both older adult groups presented with a more negative slope compared with the YA groups indicating a faster decrease of connectivity strength as the Euclidean distance increases, which was significant at a $p$-value = 0.0002, Fig. 2a, b. Differences in the intercept were found for the TFC cohort, with OAs showing a higher intercept compared with YAs ($p$-value < 0.0001) but no differences for this parameter were found for the NKI cohort. The covariate for sex was significant for the NKI group indicating differences in connectivity strength between males and females, $p$-value < 0.0001. For the TFC cohort, sex was not a significant variable ($p$-value = 0.54).

Fig. 2c shows the estimation of $Q$-$Q_{rand}$ (the modularity index difference between the participant's network modularity $Q$, and that of an equivalent random network $Q_{rand}$[6], see methods section) across a range of network edge densities for the NCL cohort, and which reached its maximum at 3.24% for the 451-ROI atlas; this edge density is referred to as optimal density (see methods section). Optimal density estimation for the other cohorts is shown in Supplementary Fig 2. When comparing the distance-vs-strength profile for the NCL cohort, all dementias showed a significant negative slope that was steeper than the OA group ($p$-value < 0.0016). The ADD group showed no significant difference for the intercept when compared with OA

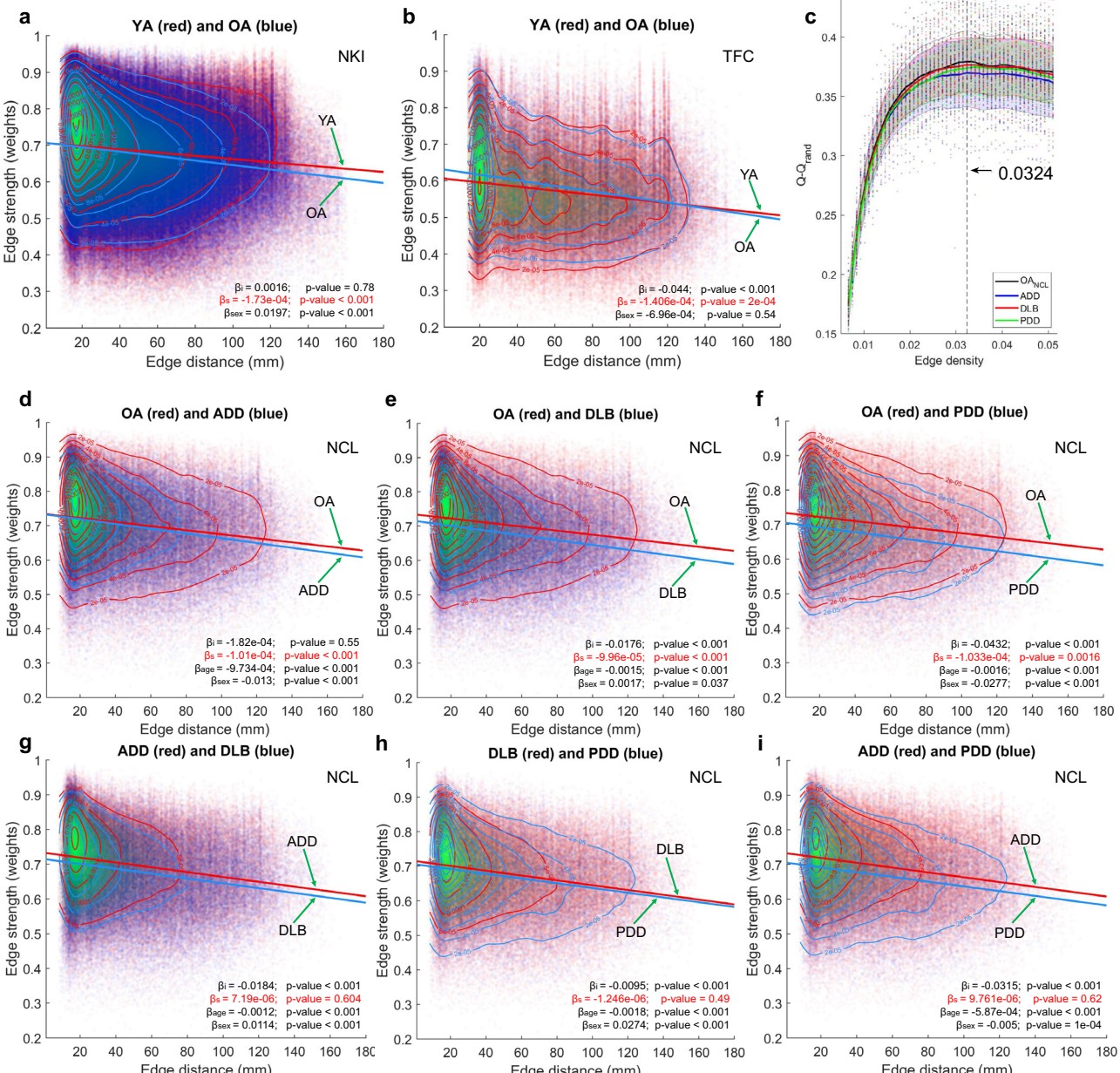

**Fig. 2 Network edge strength versus Euclidian edge distance. a** NKI; slope and intercept comparisons between edge strength and Euclidian distance profiles for the weighted thresholded connectivity matrices at optimal edge density, 3.8% with a 451-region of interest (ROI) atlas. **b** TFC; slope and intercept comparisons for the weighted thresholded connectivity matrices at optimal edge density, 5.11% with a 177 ROI atlas. **c** Optimal density estimation for the NCL cohort. Optimal edge density for all groups was reached at 3.24% with a 451 ROI atlas; see Supplementary Fig 2 for the rest of cohorts. The dashed lines and shaded error regions are the mean and standard deviation values across participants for each group, respectively. **d-f** NCL; slope and intercept comparisons between edge strength and Euclidean edge distance profiles comparing OA vs ADD (**d**), OA vs DLB (**e**) and OA vs PDD (**f**). The three neurodegenerative dementias showed a significantly steeper slope when compared with OA. **g-i** NCL; slope and intercept comparisons between dementias. The three dementia groups did not show differences in their slopes. Coefficients; $\beta_i$ = intercept difference, $\beta_s$= slope difference, $\beta_{age}$= age, $\beta_{sex}$= sex.

($p$-value = 0.55) whereas the DLB and PDD groups showed a significant lower intercept ($p$-value < 0.001). However, when comparing between dementias no significant slope differences were found, indicating that their strength-vs-distance connectivity profile was broadly similar, although their intercepts were different in all between-dementia comparisons ($p$-value < 0.001). Contrasting with the experiments for the ageing cohorts, age and sex were added as covariates of no interest. In all comparisons these covariates were significant, indicating that for the NCL cohort, age and sex influenced functional connectivity strength.

Because of this, age and sex were added as covariates of no interest in all subsequent analyses.

**Modular variability is higher in association cortices than primary cortices.** Due to the unbalanced group sizes in our study, we estimated MV using a bootstrapping approach where at each iteration an equal number of individuals from each group were randomly sampled without replacement and their community consensus was estimated (see Methods section for details on this

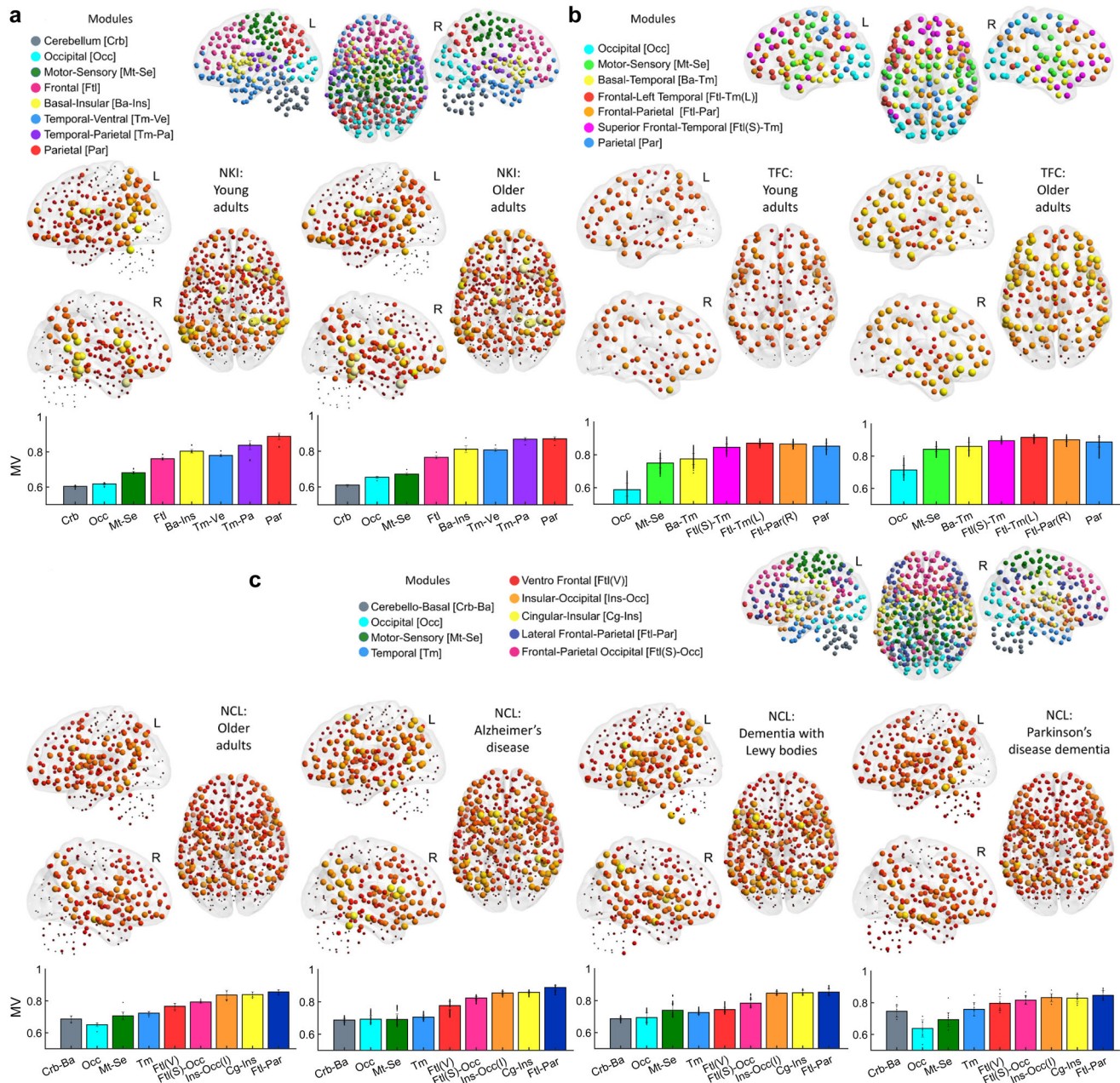

**Fig. 3 Modular variability (MV), consensus communities and group means. a** Nathan Kline Institute (NKI) consensus modularity and module definitions shown in coloured spheres (top). Group mean MV for young adult (YA) and older adult (OA) groups (middle). Mean MV per module is shown as bar plots for the OA and YA groups (bottom); colours for each bar match the communities. **b** Same as (**a**) but for the 1000 functional connectome (TFC) cohort. **c** Same as (**a**) for the Newcastle (NCL) cohort. The left hemisphere (L), right hemisphere (R). Results presented here used the 451-ROI atlas for the NKI and NCL cohorts at optimal edge density; for results using other atlases see Supplementary Fig 5. The mean values and standard deviations shown in the bar plots are obtained by 500 times with bootstrapping approach.

bootstrapping approach). Then, MV was computed as the variability between each selected participant and the consensus community at that iteration. These consensus communities were saved and a final meta-consensus was estimated and shown in Fig. 3 at the top of each panel. For the NKI cohort, eight consensus modules were found (Fig. 3a), seven modules in the TFC cohort (Fig. 3b), and for the NCL a total of nine modules (Fig. 3c). Several modules were consistent across cohorts. For instance, occipital and motor-sensory modules were found in the three cohorts. Additionally, for the NKI and NCL cohorts, which had the same parcellation and data pre-processing pipeline, cerebellar, motor-sensory, and occipital modules were also

consistent. There were some notable differences, however; for instance, nodes within the basal brain were grouped with the insula cortex for the NKI cohort, while for the NCL cohort these same nodes were grouped with the cerebellum as a community.

We then studied group modular variability (MV) in our three study cohorts. Mean MV for all groups is shown in Fig. 3a-c. All three independent cohorts and their subgroups showed a consistent pattern of MV across the brain. MV was higher at inferior frontal, parietal, insular cortices, inferior post and pre-central gyri as well as superior temporal gyri. Regions of low MV were the occipital, superior motor-sensory (or superior central gyri), temporal poles and the cerebellum (the latter for the NKI

and NCL cohorts only). The superior frontal cortex also showed low MV but to a lesser extent compared with the occipital cortex. Even though the TFC cohort database did not have cerebellar connectivity, had a lower atlas resolution (177 ROI) and an independent pre-processing pipeline, this cohort showed a similar MV pattern to the other two cohorts (Fig. 3b).

The within module mean MV for all groups is shown as bar plots at the bottom of each panel in Fig. 3. For all cohorts, the occipital and cerebellar modules showed the lowest MV values, followed by the motor-sensory modules. Modules with high mean MV were those with frontal, insular, and parietal aspects in their topological distribution; specifically, parietal and insular cortices were the regions with the highest MV.

We also estimated modular dissociation (MD). In this case, MD can be estimated within each participant (i.e. bootstrapping was not necessary) and mean MD was computed directly. Similar to MV, the pattern was consistent across the three cohorts although similarities were more evident between the NKI and NCL groups; the mean MD maps are shown in Supplementary Fig 3. For the TFC, low MD is found in occipital, and pre and post-central gyri, while high MD is found in temporal poles and one node in the ventromedial prefrontal cortex. NKI and NCL participants also showed low values of MD in occipital, and motor-sensory cortices while regions of high MD were located primarily in cerebellar, basal structures and insular cortices. Similar results were found in other atlases as well (see Supplementary Fig 6).

**Healthy ageing showed consistent patterns of increased modular variability and decreased modular dissociation**. We performed between-group comparisons for MV and MD, and for the NCL cohort, we were interested in comparisons between OA and the neurodegenerative dementias. All tests were corrected for age, sex and study by regressing out these covariates before non-parametric permutations for the between-group comparisons.

The TFC-OA group showed an overall increase of MV across the brain while the NKI-OA group showed regions of high and lower MV when compared with YA (Fig. 4a, b). However, despite this difference both ageing groups showed a similar pattern of high MV within occipital, insular and ventral frontal cortices in OA compared with YA. For the NKI groups, OA showed lower MV within parietal, motor-sensory, cerebellum and part of the superior frontal cortices (Fig. 4a).

The MD patterns for both ageing cohorts, NKI and TFC, were also similar. Again, insular areas showed significantly higher MD in older adults while cerebellum, temporal and motor-sensory regions showed lower MD in OA compared with YA (Fig. 5a). For the TFC cohort, in particular, the occipital cortex showed significantly higher MD in OA compared with YA (Fig. 5b); this trend also existed within the NKI cohort, but it was not significant after corrections for multiple comparisons. Between-group comparisons for both ageing groups survived FDR correction for multiple comparisons at a $p$-value < 0.05. Corrected results for the NKI comparisons for MV and MD are shown in Fig. 4c and Fig. 5c respectively.

**Neurodegenerative dementias show differentiated patterns of modular variability and dissociation**. When comparing dementia patients with OA within the NCL cohort, the ADD group showed on average higher MV within the occipital cortex while lower MV was found in the rest of the cortex, Fig. 6a. The DLB group also showed higher MV in the occipital cortex as well as in the motor-sensory cortex, Fig. 6b. Surprisingly, the DLB group showed regions that were statistically different, but the direction of this difference (higher or lower than OA) changed

between iterations during the bootstrapped meta-analysis, resulting in low differences (shown with pale grey colour) but significant and represented by large spheres. These regions were located primarily within the insular and ventral frontal cortices. For the PDD group, differences were not significant for MV (uncorrected) compared with OA, with only a trend of high MV within the cerebellum, Fig. 6c.

MD was also compared between dementias and OA, where the ADD and DLB groups showed a similar pattern of MD with higher values in occipital and motor-sensory cortices and lower values within frontal, ventral frontal, precuneus and basal brain regions. PDD, on the contrary, showed a differentiated pattern of MD compared with the other two dementias, with lower MD at the occipital cortex and high MD within cerebellar, insulo-opercular and motor-sensory cortices, Fig. 7.

**The functional brain moves towards a segregated modular structure with ageing whereas neurodegeneration alters this segregation locally**. We further decided to explore MD differences between young adults and the neurodegenerative dementia groups. This was possible by using the healthy OA groups within the NKI and NCL cohorts as reference groups; MD values were analysed relative to the OA groups to create an $MD_{/OA}$ index (see Methods section). Fig. 8a shows MD values from both OA groups. There was a high agreement between both OA groups regarding MD; Pearson $r = 0.8$, $p$-value = 1.17e−102, $R^2 = 0.64$. Fig. 8b shows the global MD across all nodes; here MD was significantly higher in YA compared with OA globally while this measure did not change significantly in our exemplar dementia groups indicating that globally MD is not affected by neurodegeneration; i.e. $MD_{/OA}$ was significantly different from zero in YA but not in the dementia groups. However, when we analysed MD changes relative to OA for each of the identified communities, higher YA MD was present in six of the nine modules, as shown in blue in Fig. 8c; ventral frontal, fronto-parietal, temporal, motor-sensory, cerebello-basal, and lateral fronto-parietal modules. For the modules with a frontal component, the MD change was on the average negative in the dementia groups (Supplementary Fig 4e, f and i), indicating lower MD compared with OA. For the temporal module, the ADD group showed a positive change although this was significantly lower than YA MD changes (Supplementary Fig 4c), and for the cerebello-basal module the PDD group showed on average a positive MD change, but this was significantly lower than the YA MD change. For the motor-sensory module, all groups showed a positive change relative to OA, but this change in dementias was significantly lower than YA (Fig. 8c, Supplementary Fig 4a). From these results, it is worth noting that for the occipital module, the PDD group showed a significant negative change of MD which contrasted with the positive MD change in ADD and DLB groups (Supplementary Fig 4g).

## Discussion

Our research study showed consistent patterns of group modular variability (MV) and modular dissociation (MD) across the three independent neuroimaging cohorts and which suggest pattern universality for these two characteristics. Our main findings are that in the ageing brain, its modules move towards a connectivity regime that favours segregation while brain modules become more heterogeneous in older adults compared with young adults. This shift towards segregated modular connectivity is not affected by neurodegeneration globally. However, at the modular level, there are marked changes where modules showed increased and decreased modular dissociation depending on the type of neurodegenerative disease.

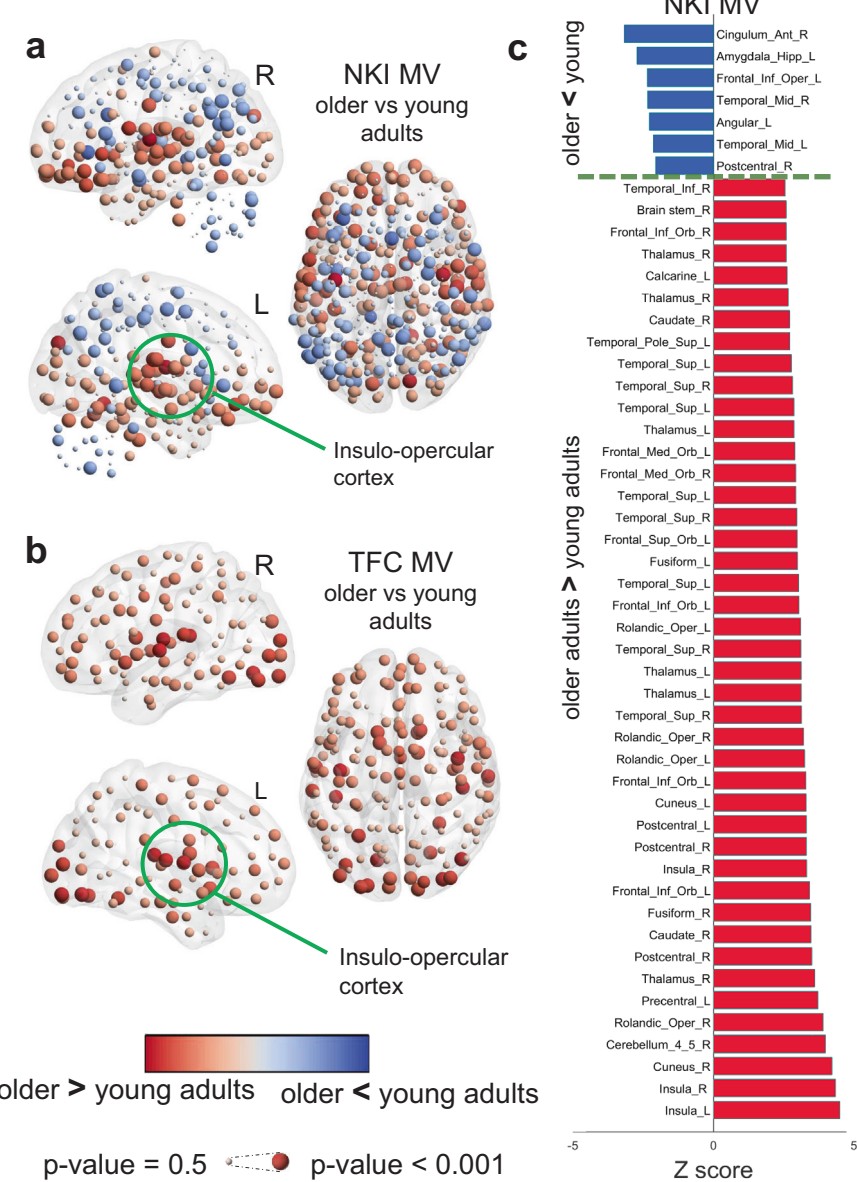

**Fig. 4 Age effects of modular variability (MV). a** Age differences in MV within the Nathan Kline Institute (NKI) cohort, older adults (OA) compared with young adults (YA). **b** Age differences in MV within The 1000 functional connectome (TFC) cohort OA compared with YA. **c** Significant brain regions from the NKI-MV comparison; corrected for multiple comparisons at p-value < 0.05. Differences were assessed with non-parametric permutations (5000) after regressing covariates of no interest. Regions' names are given using the Automatic Anatomical Labelling (AAL).

Our method of modular variability measures the variability in community assignment of a node when compared against the group's consensus community. Hence, MV measures modular heterogeneity across participants. This measure was observed to be high within association cortices: the posterior and anterior association cortices, which span the parietal and frontal lobes. The limbic association cortex which spans the temporal pole did not demonstrate high MV.

In contrast, the insular cortices showed high MV in all groups. The brain regions showing high MV are known brain areas considered of high demand; i.e. that these engage with multiple modules or brain systems and are involved in multiple cognitive tasks[7]. Indeed, the regions of high cognitive activity identified by Bertolero, et al.[7] are remarkably similar to our pattern of high MV across participants. Hence, our MV pattern maybe capturing the different interactions between nodes of high demand that connect with a great variety of regions and modules.

Brain regions with low MV were the motor-sensory, occipital cortex and cerebellum. The motor-sensory and occipital cortices are known primary cortices that receive information from thalamic connections[30]. These network modules are also the most consistently reported in the literature indicating that these are highly integrated into the functional brain and that nodes within these modules interact less often with other brain regions[31]. The cerebellum was found here to be a module with low MV, this structure is highly connected to the basal ganglia[32] and it is topologically isolated from the cortex, with the dentate nucleus as its main route to the basal brain[33] and which may drive its consistent modular integrity.

A hypothesis for the differences in modular variability between the association and primary cortices is that during the neurogenesis, the association cortex develops later in humans allowing flexibility of function and behaviour[34], whereas primary cortices develop earlier with strong connections from the basal brain,

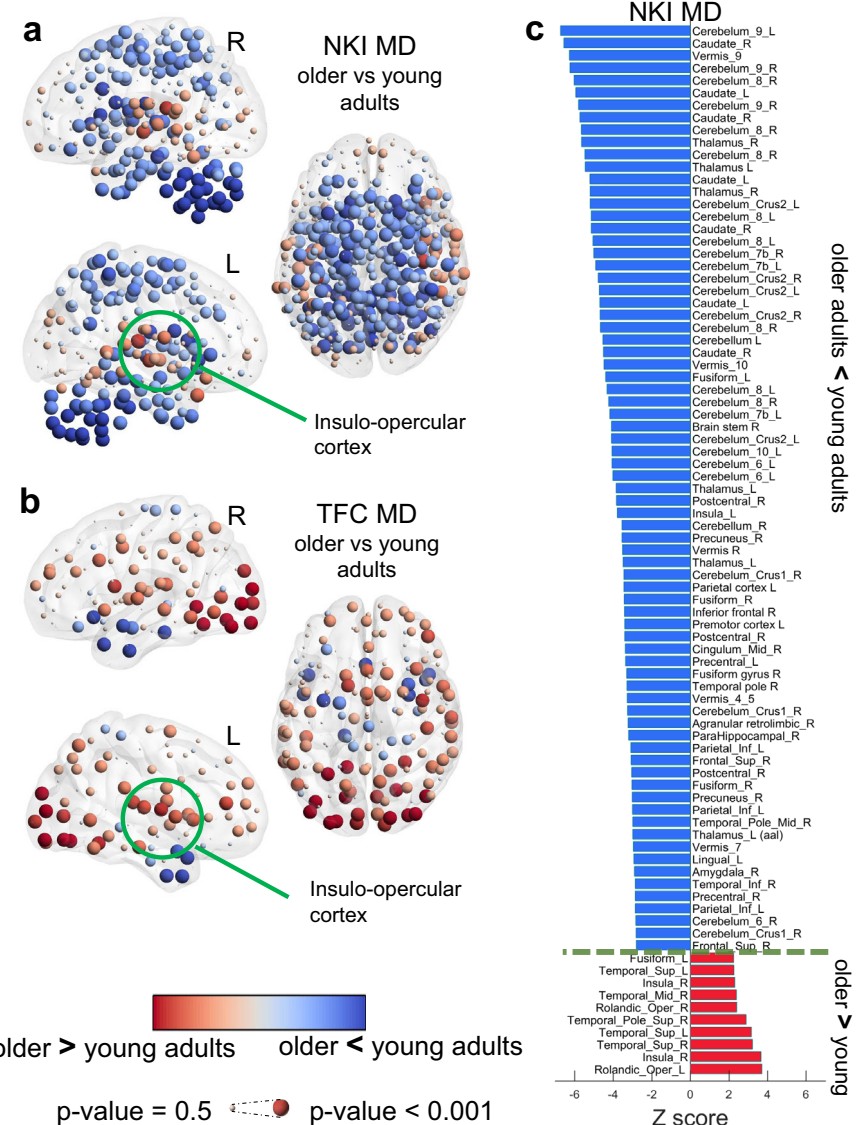

**Fig. 5 Age effects of modular dissociation (MD). a** Age differences in MD within the Nathan Kline Institute (NKI) cohort, older adults (OA) compared with young adults (YA). **b** Age differences in MD within The 1000 functional connectome (TFC) cohort OA compared with YA. **c** Significant brain regions for the NKI-MD comparison; corrected at *p*-value < 0.05. Differences were assessed with non-parametric permutations (5000) after regressing covariates of no interest. Regions' names are given using the Automatic Anatomical Labelling (AAL).

which may also control their cytoarchitecture and functionality[35]. The high modular variability observed in particular within the insular regions and operculum seems to indicate high demand. The insula in the human brain is characterised by the presence of von Economo neurons, which are fast communication circuits within the brain[36]. Additionally, it has been reported that the insula is involved in multiple brain functions, regulating salience stimuli and activity between brain systems such as the attention and default mode systems[37].

The patterns for MD and MV showed low values within the motor-sensory and occipital cortices/modules indicating that at the group level these modules are very homogeneous across participants and are densely connected within their respective communities, i.e. segregated by nearest neighbour connectivity.

The MD pattern showed remarkably high values within basal regions in all groups. Because of the definition of MD, this index measures the difference between global and local threshold construction methods. This difference is mainly driven by the weakest connections introduced by the local thresholds and

which change the topography of communities. These connections are not weak in correlation given that these belong to the strongest nearest neighbours for each node within the participant's connectivity matrix. However, the higher MD values within the basal region indicate that nodes within this region have a high tendency to disconnect or join other modules, i.e. dissociate. In this regard, the basal brain and specifically the thalamus presents with weak and sparse connections to the cortex[30], even though these connections can control the global functional dynamics of the whole cortex[38].

The structure of weak connections in the brain has been a matter of intense research. These connections are found to be non-random, and their function is thought to relate to network path length shortening[39]. Also, previous research has found that the weak long-distance connections contribute to the brain's complexity[5] and that their strength decreases with cortical distance[40], a finding that we also confirmed here (in Fig. 2). Previous investigations in modular connectivity have used the strongest connections using either static[10] or dynamic[16]

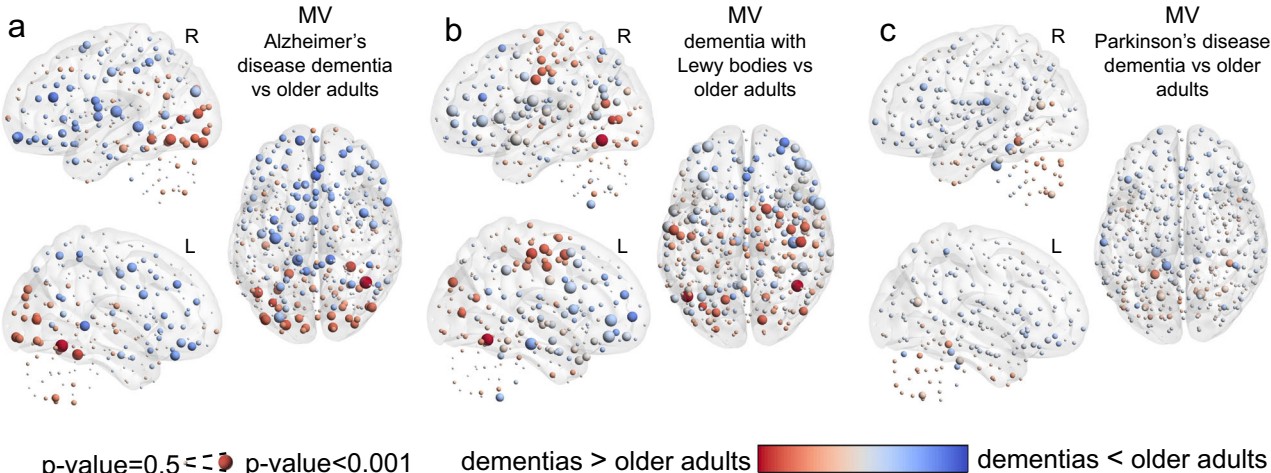

**Fig. 6 Neurodegenerative dementia effects of modular variability (MV). a** MV differences between older adults (OA) and Alzheimer's disease dementia (ADD). **b** MV differences between older adults (OA) and dementia with Lewy bodies (DLB). **c** MV differences between older adults (OA) and Parkinson's disease dementia (PDD). Results were assessed with non-parametric permutations (5000) after regressing out covariates of no interest. Results for MV-PDD were not significant (uncorrected) while the rest of the comparisons showed significant uncorrected results.

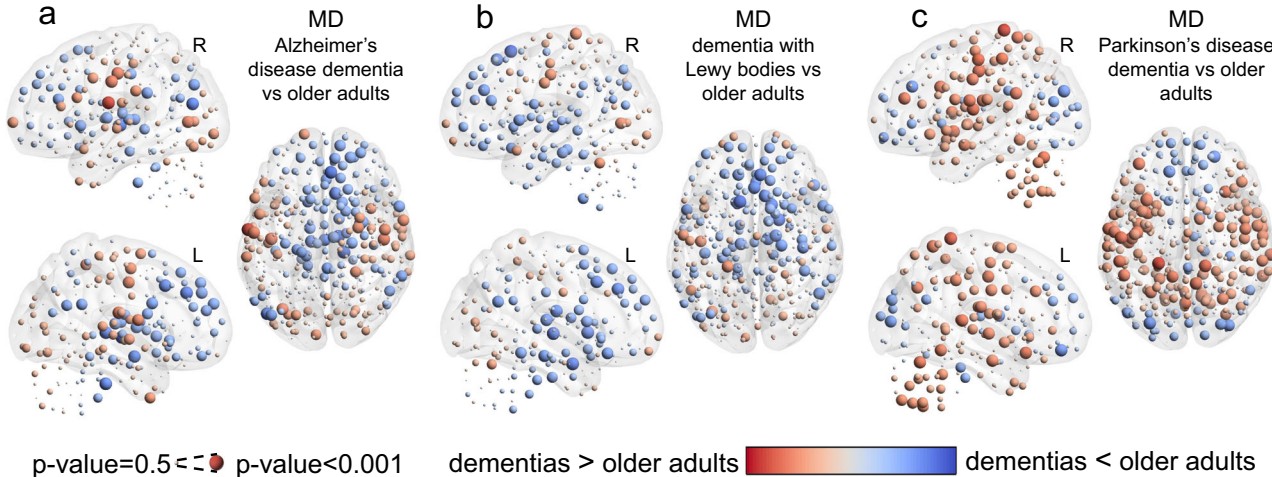

**Fig. 7 Neurodegenerative dementia effects of modular dissociation (MD). a** MD differences between older adults (OA) and Alzheimer's disease dementia (ADD). **b** MD differences between older adults (OA) and dementia with Lewy bodies (DLB). **c** MD differences between older adults (OA) and Parkinson's disease dementia (PDD). Results were assessed with non-parametric permutations (5000) after regressing out covariates of no interest. Only results for MD-PDD survived correction for multiple comparisons at a *p*-value < 0.05.

approaches. It is highly plausible that because of the weak and dynamic nature of the basal connections, these previous studies were not sensitive to basal connectivity, even though these are highly targeted by a wide range of brain diseases and conditions[41].

The comparisons between ageing groups, OA vs YA, from both cohorts showed similar patterns of MV and MD suggesting that the ageing process follows a consistent path of alterations. Also, our results demonstrate that generally, the brains of OA become more segregated with lower dissociation values compared with YA. The motor-sensory, temporal cortex and cerebellum which are regions of low MD, have this metric further decreased in OA while regions within insular and occipital cortices show increased MD. It is hypothesised that modular segregation is a result of functional specialisation[17]. In this regard, Baum et al.[17] found, using structural networks from diffusion tensor imaging (DTI), that modules segregate from childhood to adulthood, which agrees with our results using functional connectivity and further extrapolates this conjecture to older adults. Baum et al. also

reported that the opercular cortex does not follow segregation with age which agrees with our result in functional networks. The high MV and MD shown by the OA group in the insulo-opercular cortex compared with YA were observed in both NKI and TFC cohorts indicating a universal characteristic.

The occipital cortex also showed higher MD and MV in OAs from both ageing cohorts, although this characteristic was more prominent in the TFC cohort. The occipital cortex is one of the most resilient regions to ageing; cortical thickness in this region does not significantly decrease with ageing[42] and shows minimal thinning/atrophy in ADD[43]. However, this cannot explain the higher MD and MV observed in OA because cortical thickness within the insular cortex, contrary to the occipital cortex, is significantly reduced by ageing[42]. At this point, our only explanation is that both regions are of higher demand in the old age brain.

When we compared MV and MD between dementias and older adults, only results for the comparison PDD vs OA survived correction for multiple comparisons. This also indicated that for ADD and DLB, our two nodal indices remained fairly invariant to the

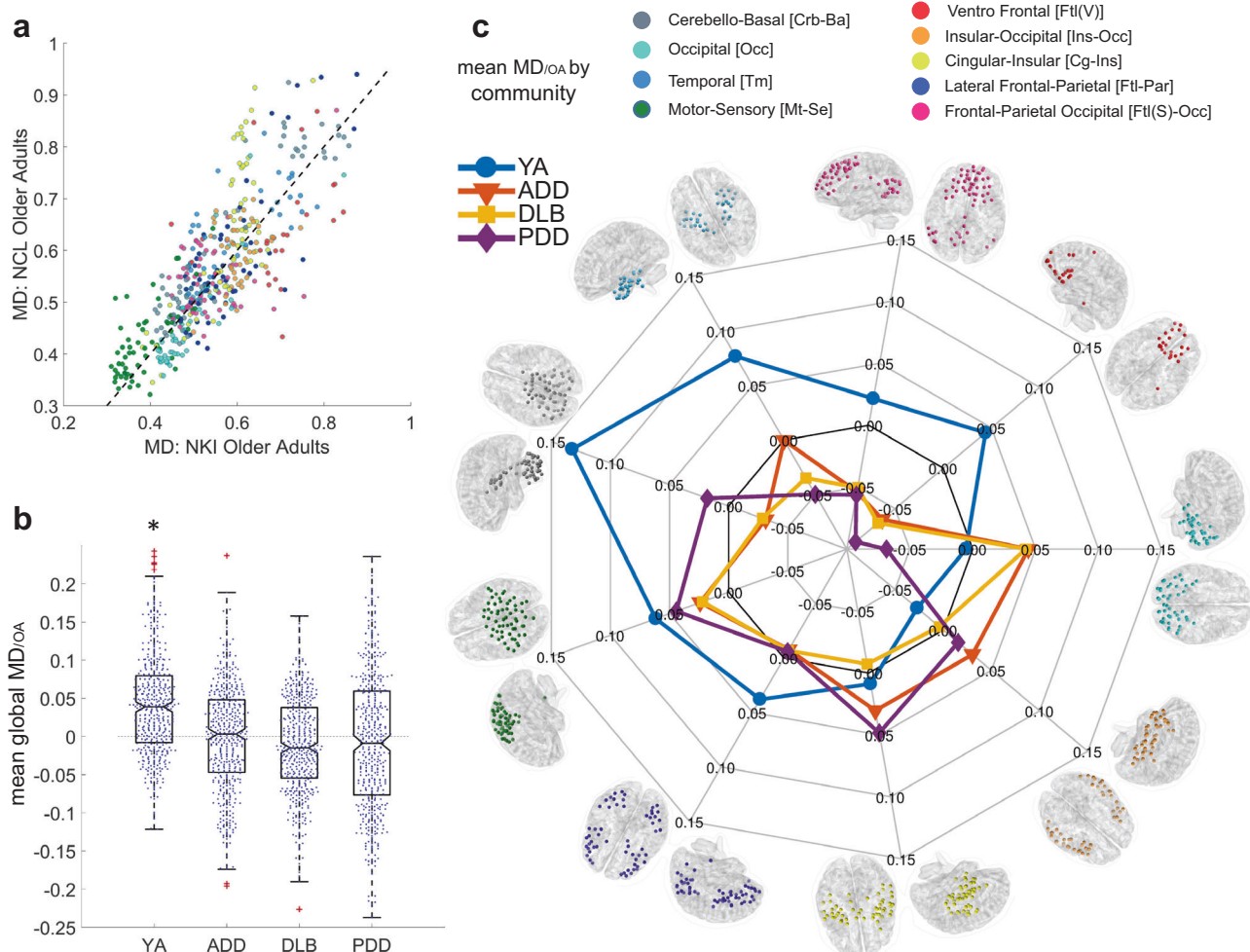

**Fig. 8 Modular dissociation relative to older adults, MD/OA; changes from healthy ageing to dementia. a** Scatter plot with MD values from the OA groups within the Nathan Kline Institute (NKI) and the Newcastle University (NCL) cohorts; colours are shown according to communities. Both OA groups were correlated; Pearson $r = 0.8$, $p$-value $= 1.17e{-}102$, $R^2 = 0.64$. **b** Global $MD_{/OA}$ for the NKI-young adult (YA) group and the NCL groups; Alzheimer's disease dementia (ADD), dementia with Lewy bodies (DLB), and Parkinson's disease dementia (PDD). *Statistically different from zero, Wilcoxon signed-rank test: $p$-value $= 1.183e{-}29$. The mean values and standard deviations are derived from the whole brain across the nodes. **c** Spider plot for Mean $MD_{/OA}$ by the community. Ageing drives a significant decrease of MD shown by the high scores in the YA group (blue line) and mainly for the cerebello-basal community, suggesting that the ageing brain moves towards a local connectivity regime of segregation. Neurodegenerative dementia did not change the overall brain's $MD_{/OA,}$ but alterations are present in different communities. See Supplementary Fig 4 for non-referenced results of MD values and box plots.

presence of these neurodegenerative dementias. However, a pattern was observed in ADD and DLB; these dementias showed a trend of higher MD in the occipital and motor-sensory cortices. This contrasted with the NKI and TFC cohorts where OA showed lower MD for these regions compared with YA, which suggests that MD in the motor-sensory cortex partially returns to YA's levels in the presence of neurodegeneration and that this region departs from a modular segregation regime that is a characteristic of ageing.

PDD showed higher MD in motor-sensory and insular cortices as well as in the cerebellum; only occipital and frontal cortices showed a trend that favours segregation of modules. These differences may be related to the neuropathology of the Lewy body diseases affecting motor and cognitive brain functions. In a recent investigation by Kim et al.[44], the authors reported an increase in network state transitions, which they called State I and II networks, in PD patients and which suggested unstable brain activity. This agrees with our finding of a higher MD in our group of PDD patients and which indicates a departure from segregated but structured brain modularity. However, more research on larger cohorts will be needed to bring more light into this.

The global mean MD across the cortex did not change significantly with the presence of dementia, which is particularly visible in DLB and PDD, contrary to normal ageing which showed lower MD in OA compared with YA. However, and despite this global invariance, at the modular level, there were significant changes in all dementia groups. Six of the nine modules showed lower relative MD than the YA group. This first indicated that with ageing, the brain is more segregated, with nodes closer in relative strength to their communities. However, the effects of dementia vary, for some modules, the tendency is of higher or lower MD depending on the type of dementia but in many of these cases this was significantly different from zero, indicating the change. For instance, the negative relative MD in all dementias for the ventro-frontal module (Fig. 8c: the module in red) shows that during ageing, nodes within this module follow a connectivity regime that favours modular segregation, and that in the presence of neurodegeneration, nodes follow this tendency even further (negative $MD_{/OA}$). The contrary occurred with the motor-sensory module, where the positive relative MD in all dementias shows that this module is partially 'restored' to young

adult levels of dissociation. The pattern of $MD_{/OA}$ between ADD and DLB is very similar, confirming the similarities between these two diseases. Surprisingly, the PDD group showed a completely different $MD_{/OA}$ pattern compared with DLB, even though previous investigations have suggested functional similarities between DLB and PDD[25,45].

Previous investigations have focussed their attention on the strongest connections. This is certainly sensible; the strongest connections are less influenced by noise and are less likely to be of artefactual origin. However, recent investigations have shown the importance of weak or slightly weaker connectivity within the brain. In a resting state, fMRI investigation linking brain connectivity with intelligence quotients (IQs) from participants, Santarnecchi, et al.[46] found that the 60–80% range of the weakest end of the edge-weight distribution, correlated with IQ, and this correlation was higher and more significant than for the strongest connections at the 1–20% range. In our investigation, we performed a test to observe where in the connectivity matrix the weak edges were added by the local thresholding method (Supplementary Fig 7). Most of the added edges by the local threshold construction method were intra-modular edges, i.e. these edges made communities concise. On the contrary, edges added by the global threshold were inter-modular, or in other words these aimed to fuse modules; especially the motor-sensory and temporo-parietal modules, Supplementary Fig 7c, g. Additionally, the number of intra-modular edges (by their mean node degree from binarised edges) added by local thresholding correlated positively with global MD, proving their influence on the MD index.

The network construction method by local thresholding showed interesting properties in our investigation. Using this method, the modularity statistic Q was not different between dementia groups and OA, and the same can be seen for the NKI database of OA and YA, Supplementary Fig 2. Also for both cohorts, Q variance is lower when using the local thresholding method. Another observation is that the mean number of modules estimated by local threshold is the same for all groups within each of the databases (NCL = 9, TFC = 7, and NKI = 8 modules at their respective optimal edge densities). This is because the local thresholding method is based on the $k$-nearest neighbour graph, which favours local connectivity by connecting the closest nodes and inducing a more segregated network. This was confirmed by analysing the edge Euclidean distance distribution for the connections introduced by the local threshold network construction regime, where close-range connections are highly introduced. Also, the long-range weaker connections, although less probable than short-range, are more probable than in a global threshold regime (Supplementary Fig 8). In conclusion, the weaker connections added by local thresholding, and which connected nearest neighbours, did not aim to communicate between modules but to consolidate modules, even in the presence of disease, in our case, neurodegenerative dementia.

Fig. 9 shows an explanation for the introduction of short-range weaker connections by the local thresholds in neurodegeneration. Currently, it is accepted that neurodegenerative diseases target specific brain systems or modules;[47] neurodegenerative diseases affect connectivity within these specific systems and their short-range connections[48]. Despite being affected by neurodegeneration, these local links are still connecting the nearest neighbours within their respective communities and can survive a local connectivity regime (Fig. 9c). The finding that modularity statistic Q (Supplementary Fig 2) and the number of modules were not different between dementias and OA using local thresholds support this.

Recent studies have suggested that head motion influences functional connectivity measures[49–51]. Although we performed

motion correction in the pre-processing steps of resting-state fMRI images before analysis, the impact of small movements is still uncertain. We thus estimated the framewise displacement (FD, a motion index that integrates the six motion parameters inferred by FSL-FLIRT) for each participant and removed participants whose mean FD was larger than 0.5 mm. Supplementary Fig 9 showed mean FD values for all participants and groups in NKI and NCL cohorts. And the study demographics after exclusion of large mean FD was shown in Supplementary Table 2. The mean FD was then added as a covariate of no interest together with sex and gender in the repeated analysis to test if the motion affected our results or not. Only the NCL and NKI cohorts were studied since the TFC cohort was pre-processed in a previous study and no motion parameters were available[22]. As shown in Supplementary Fig 10, the motion indeed influenced the functional connectivity strength, particularly affecting the short-range connections, which was consistent with the previous work[51] showing increased/diminished connectivity for short-/long-distance edges. However, the motion was found not to affect our main results. The related details and results (Supplementary Fig 11–16) are described in the Supplementary Discussion. Identical to the analysis without FD regression, the association cortices were found to have larger MV than primary cortices. The mean MD followed similar patterns to MV with low values within the motor-sensory and occipital modules. Compared with the YA group, the OA group showed high variability and was more segregated with lower dissociation values. Although the modules and MD patterns within modules varied for the NCL cohort, the three dementia groups were also shown not to differ in terms of global mean MD compared to the OA group. And compared to OA and dementia groups, the YA group consistently showed higher MD which was mainly distributed in frontal related modules, temporal cortex and cerebellum. This group-level regression analysis of FD was widely used to correct motion artefacts[51], however, in addition to this noisy effect, the neural effect of head motion has been previously highlighted[50], which may lead to changes in neural activity and thus to be one limitation of our work.

In addition to the potential effect of head motion, it also remains unclear if the findings are robust on the basis of well-established brain parcellations. In order to investigate this further, we conducted a replication study using a multi-modal atlas Human Brainnetome Atlas[52] which in total included 210 cortical, 36 subcortical and 28 cerebellar subregions. The mean FD, sex and age were added as covariates of no interest in analysis. The corresponding results were shown in the Supplementary Discussion and Supplementary Fig. 17–22. The modular definitions at the optimal edge density of 4.40% varied compared to the edge density of 3.24% based on 451-ROI atlas. For example, the frontal-parietal module in the 451-ROI case changed to be two separated modules for the NKI cohort, while two separate modules in the 451-ROI case: cerebellum and occipital modules were integrated into one module for the NCL cohort. Despite modular differences, the general MV and MD distribution of low and high values were maintained. When comparing OA and YA groups, the patterns of high MV and low MD for the OA group were weakened and even diminished on the standard atlas. However, the features of both high MV and MD in the insulo-opercular cortex, the trend of low MV and MD in parietal, cerebellum cortices were comparable. Similarly, healthy ageing moves towards a segregated modular structure from a young age and mean global MD in DLB and PDD groups did not differ significantly from the healthy ageing group. The ADD group, on the contrary, was found to have higher mean MD than OA, which was not shown in the 451-ROI case. The differences between standard parcellation and in-house parcellation analysis mainly

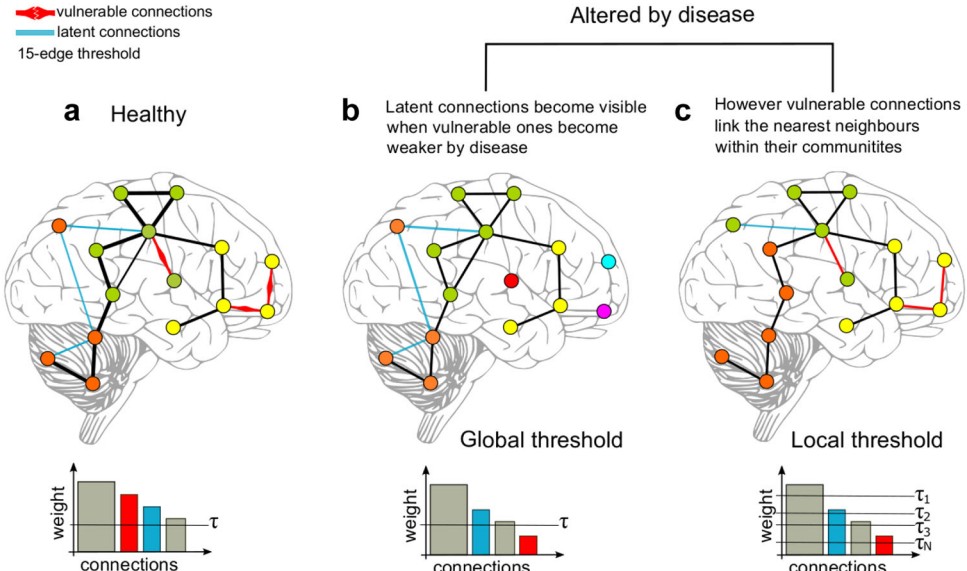

**Fig. 9 Disease effects on local and global thresholding. a** A toy model of healthy brain connectivity with "known" connections including vulnerable edges which can be targeted by disease, and latent connections which are weaker than vulnerable connections. **b** When the model is targeted by disease, the vulnerable connections become weaker and do not survive a global threshold. **c** Although weakened by disease, the vulnerable connections still connect the nearest neighbour nodes within their communities, and these connections are restored or made visible by local thresholds.

existed in the MV comparison between YA and OA (Supplementary Fig 19a), and the mean MD comparison between ADD and OA (Supplementary Fig 21b). The former differences could be caused by the bias of registration between standard parcellation and our study cohorts, which underwent different MRI scans and were acquired at different study sites. Another factor could be the resolution of the Human Brainnetome Atlas including 274 ROIs compared to the 451 ROI in-house atlas, which may weaken or decrease the significance because of the global mean across the region. The large similarity of decreased MV in the OA group between the results on Human Brainnetome Atlas and in-house 247-ROI atlas (Supplementary Fig 19a, b vs. Supplementary Fig 19c, d) supports this. The later differences, on the other hand, could be owing to the confounding effect of head motion and the standard atlas. Evidence has been shown in the following: the global mean $MD_{/OA}$ for ADD group increased to be positive after regressing out head motion even though this was still not significant (Supplementary Fig 15b vs Fig. 8b); the use of the low-resolution Human Brainnetome Atlas promoted a further increase of the mean $MD_{/OA}$ for ADD group which turned to be significant (Supplementary Fig 21b). Overall, the standard parcellation was demonstrated to affect the comparison results definitely but the primary findings generally remained, especially for the MV/MD distribution patterns, the high MV/MD of OA group in the insulo-opercular cortex compared with YA, and local but not global changes of neurodegenerative dementia groups: DLB and PDD groups, which help convinced the previous studies based on 451-ROI atlas.

In conclusion, high modular variability or heterogeneity of modules exists within the association cortices which are known regions of high cognitive demand. Our results on the 451-ROI atlas showed that this is a universal characteristic of the brain, whose changes are shown to be consistent in healthy ageing. Similarly, modular dissociation was consistently high at basal brain regions, and this brain characteristic decreases with healthy ageing, indicating that the brain moves towards a connectivity regime that favours segregation of modules, except for the insulo-opercular cortices which depart from this regime. In contrast, modular dissociation is not affected by neurodegenerative

dementia globally but at the modular level, which is particularly visible in DLB and PDD. The brain is constantly learning and also gets experienced. The ability of the brain to dissociate modules in youth may be a reflection of continuous learning and the need for neuronal modules to communicate and share resources. As the brain gets older, these interactions get impaired by ageing and the brain moves towards a more segregated network structure. Despite this, brain modules in older adults are more heterogeneous across participants suggesting that the brain within individuals also follows different strategies or paths during the ageing process.

## Methods
### Experimental model and participant details
*Newcastle participants and recruitment.* Two independent neuroimaging databases were combined from two clinical studies[23–25]. Participants in these studies were recruited within the north-east of England and patients with neurodegenerative dementia were contacted through old-age psychiatry and neurology services in the Newcastle area (United Kingdom). A total of 42 dementia with Lewy bodies (DLB, $N = 16$ in study 1 and $N = 24$ in study 2), 44 Alzheimer's disease dementia (ADD, $N = 16$ in study 1 and $N = 28$ in study 2) and 17 Parkinson's disease dementia (PDD, $N = 17$ in study 2 only) patients were recruited. Additionally, 34 age-matched healthy control participants ($N = 16$ in study 1 and $N = 18$ in study 2) were recruited as a comparison group. All patients were diagnosed by two experienced clinicians according to the clinical criteria for these diseases: Dementia with Lewy bodies consensus criteria[53,54], the diagnostic criteria for PDD[55], and the National Institute on Aging-Alzheimer's Association criteria for AD[56]. For both studies, approval was granted by the Newcastle Ethics Committee and all participants gave informed consent.

*Enhanced Nathan Kline Institute (NKI) Rockland Sample.* Participants from the NKI-Rockland sample[21] were selected and neuroimaging datasets downloaded. Selection criteria comprised participants with resting-state (fMRI) who were between 20–40 years old for the young adult group (YA, $N = 151$) and participants between 60–80 years old who comprised the older adult group (OA, $N = 151$). Details about the recruitment of participants for this cohort can be found in Nooner, et al.[21].

*The 1000 functional connectome project.* Functional connectivity matrices from the 1000 functional connectome[22] were downloaded from the USC Multimodal Connectivity Database (UMCD, http://umcd.humanconnectomeproject.org/)[57]. This connectivity repository comprises resting-state connectivity matrices (Pearson correlations) from nine independent studies with a wide age range. A young adult group (YA, $N = 257$ participants between 20–40 years old) and an older adult

group (OA, $N = 102$ participants between 60–80 years old) were selected and functional connectivity matrices were downloaded.

### Neuroimaging procedures

*Neuroimaging acquisition from Newcastle participants.* Neuroimaging was acquired in both studies with the same scanner. Structural magnetic resonance imaging (MRI) was recorded with a 3 T Philips Intera Achieva Scanner. Acquisition protocol used a magnetisation prepared rapid gradient echo (MPRAGE) sequence, sagittal acquisition, echo time 4.6 ms, repetition time 8.3 ms, inversion time 1250 ms, flip angle 8°, SENSE factor = 2, and an in-plane field of view of $256 \times 256$ mm$^2$ with a slice thickness of 1.2 mm yielding a voxel size of $0.93 \times 0.93 \times 1.2$ mm$^3$ for study 1. For study 2, an in-plane field of view of $240 \times 240$ mm$^2$ with a slice thickness of 1.0 mm yielding a voxel size of $1.0 \times 1.0 \times 1.0$ mm$^3$ was used. For functional resting-state neuroimaging, both studies used the same recording protocol: gradient echo echo-planar imaging sequence with 25 contiguous axial slices, 128 volumes, anterior-posterior acquisition, in-plane resolution = $2.0 \times 2.0$ mm$^2$, slice thickness = 6 mm, repetition time = 3000 ms, echo time = 40 ms, and field of view = $260 \times 260$ mm$^2$.

Neuroimaging protocols for the NKI and TFC databases can be found in Nooner, et al.[21] and Biswal, et al.[22] respectively and references therein.

*Neuroimaging pre-processing.* Neuroimages from the NCL and NKI databases were pre-processed with the same pipeline and blinded to the study groups. Non-brain tissue was stripped with the brain extraction tool (BET) from the FMRIB Software Library (FSL version 5.0)[58] and results were visually inspected to check if brain structures were complete and isolated. Resting-state functional MRIs (rs-fMRI) were then processed. For the NKI resting-state neuroimages, the first five volumes were deleted because the time series were not steady. All rs-fMRIs were motion-corrected using the FMRIB's Linear Image Registration Tool (FLIRT) without spatial smoothing. Then, the six-movement variables from FLIRT and the average time series from the bilateral ventricle were regressed out from all resting-state images. In order to further correct for movement and other artifacts, the rs-fMRI time series were despiked with the BrainWavelet Toolbox[59], which is a data-driven motion algorithm that subtracts high and low-frequency motion-related events. Structural and functional images were then coregistered using FEAT, and nonlinear coregistration to MNI152 space was implemented with FMRIB's Nonlinear Image Registration Tool (FNIRT) within FSL. Then, fMRI time series were high-pass filtered with a 150 s filter using FSL-FEAT. Finally, all functional images were transformed into a $4 \times 4 \times 4$ mm$^3$ resolution, and a 6-mm full-width half maxima (FWHM) spatial smoothing was applied to all volumes.

Four functional brain parcellations or atlases were estimated with the pyClusterROI toolbox:[29] ≤ 100, ≤ 200, ≤ 250, and ≤ 500 regions of interest (ROI) were tested. For this, an independent dataset of healthy older adults ($N = 29$ participants, 16 males and 13 females, recruited within the north-east of England and who had neuroimages recorded using the same scanner as the NCL cohort; mean age 64.1 (standard deviation ± 7.92), and mean MMSE, mini-mental state examination, of 29 with a standard deviation ± 0.88), were pre-processed with the same pipeline as the NCL and NKI databases. The functional brain from the independent group was parcellated with the pyClusterROI software using a grey matter mask from the Harvard-Oxford atlas (from FSL) at a 0% threshold, and which included the cerebellum. Details for the independent older adult database can be found in Schumacher et al.[23] and Peraza, et al.[60]. The functional atlases estimated and used in this study comprised 100, 200, 247, and 451 ROI. The average time series within each ROI was extracted from all rs-fMRIs, and Pearson correlation matrices were computed for each participant from the NCL and NKI databases. The main manuscript shows results using the 451-ROI atlas while results using the other functional atlases are given in the supplementary material.

The TFC matrices comprised Pearson correlations between 177 ROIs from a functional atlas[29] and had no cerebellar structures; only the neocortex and subcortical structures were included. Further details about the pre-processing pipeline used to estimate these matrices can be found in Biswal, et al.[22].

After neuroimaging pre-processing, two AD and four DLB participants from the NCL and five OA participants from the NKI cohort were excluded due to coregistration inaccuracies with the functional atlas; these older adult participants presented with structural alterations in their brains that could not be normalised to the standard MNI space.

*Community and modularity statistic estimation.* At this stage, there are three sets of connectivity matrices with Pearson correlation coefficients: NCL, NKI and the TFC cohorts. Currently, there is no consensus about the treatment of negative values from Pearson correlation matrices for the definition of network connections, and several approaches have been proposed such as deletion of all negative correlations[61], transformation to an all-positive scale[62] or compute the absolute value of the correlation coefficients $|r|$[19]. Here we decided to take the absolute value of the connectivity matrices as we did in our previous study[19] because we were interested in the connectivity strength between regions rather than in the direction of these correlations. Furthermore, previous evidence from functional connectivity studies has proved the biological origin[63,64] and the importance of negative correlations in the brain[65], further confirming our decision of keeping this connectivity information.

Networks were represented as binary undirected connectivity matrices. For this, the weighted matrices need to be thresholded and binarised. The thresholds were selected by edge density also known as network connectivity cost or proportional thresholding[15]. Network cost is defined as:

$$2T/(N^2 - N) \times 100\% \tag{1}$$

where $T$ is the total number of edges that survived the threshold and $N$ is the number of nodes. Three network edge densities were studied: optimal edge density (defined below), 10 and 20% of the strongest connections.

Modularity statistics were estimated using weighted thresholded matrices. For this, the Hadamard product between the binarised matrix and the original weighted matrix was computed, and used for community structure and modularity statistic estimation using the Louvain's algorithm function from the Brain Connectivity Toolbox (BCT)[2] in Matlab (Mathworks Inc, R2017a). In all Louvain community estimations, the algorithm was run 1000 times and the optimal community structure with the highest modularity statistic $Q$ was saved for further analysis.

*Network construction methods.* When thresholding connectivity matrices by edge density, there are two possibilities on how this threshold is applied in order to build the binarised adjacency matrix: local[12] and global threshold[66]. In the global threshold approach, a single threshold $\tau$ is applied to the entire weighted connectivity matrix selecting a percentage (cost) of the strongest edges; connectivity weights below a threshold $\tau$ are set to zero and above $\tau$ are set to one. The local threshold on the other hand, imposes a minimum node degree in all nodes within the network. Local thresholding is based on the $k$ nearest neighbour graph (K-NNG) where every node is connected to its $k$ nearest (strongest) neighbours[67]. Local thresholding works by searching the closest lower K-NNG, closest to the desired edge density, and the method adds extra edges to each node until the desired cost or the next K-NNG is reached[12]. Software to construct networks by local thresholding is provided in a public repository as supplementary material, see the data and code availability section.

As a consequence, the local threshold allows the inclusion of weaker connections in order to maintain the rule of $k$-nearest neighbours and favours integration of communities and segregation of modules within the network. Indeed, a previous investigation noted that constructing networks by local threshold enhances the identification of communities[68]. Nevertheless, the weaker connections added by the local threshold construction regime would not be present when using a global threshold approach, and the structure of both binarised networks will differ mainly at these weaker connections[15]. A toy example for the differences between local and global threshold construction methods is shown in Supplementary Fig 1 and a neurophysiological interpretation is shown in Fig. 9.

*Optimal cohort edge density selection.* The optimal cohort edge density was selected as the density that maximised the difference between the participant's network modularity and an equivalent random network modularity statistic, $Q - Q_{rand}$[6]. The modularity of the equivalent random network was estimated as the average modularity statistic of 40 randomised thresholded weighted matrices using the BCT function randmio_und_connected.m. A range of densities for each participant (from the NCL, NKI, and TFC cohorts) and atlas parcellations (100, 200, 247, and 451 ROIs) was explored for optimal edge density. From 0.3–25% (with 100 equally spaced increments) for the 100-ROI parcellation, from 0.2–10% (with 73 equally spaced increments) for the 200 and 247-ROI parcellations, and from 0.08–5% densities (with 51 equally spaced increments) for the 451-ROI parcellation. For optimal edge density discovery, we opted for Newman's algorithm[69] instead of Louvain's because the former is faster to compute, and at this step, we were not interested in the community structure per se but the edge density where the modularity statistic is maximally higher when compared with an equivalent random network. Once the optimal density was found, Louvain's algorithm was implemented instead.

*Connection length and strength comparisons.* In order to test for connectivity differences in Euclidean distance and strength profile between groups, all weighted connections from the locally thresholded matrices at optimal edge density (see the previous section on optimal edge density) were investigated with multiple linear regression models. For each group comparison, a linear model to test for differences in the intercept and slope of the distance-vs-strength regression lines between the two groups was implemented. This model was defined as:

$$strength \sim \beta_1 \cdot distance + \beta_2 \cdot group + \beta_3 \cdot group * distance \\ + \beta_4 \cdot age + \beta_5 \cdot sex + \beta_6 \cdot study + 1 \tag{2}$$

where the coefficients $\beta_2$ and $\beta_3$ account for group differences in intercept and slope respectively. However, because the correlation values within the weighted connectivity matrices $|r_{ij}|$ did not follow a normal distribution, bootstrapped linear models were implemented instead[70]. At each iteration, 20% of the total distance-strength connection population was randomly sampled without replacement together with their respective covariates for age, sex and study/site (if applicable). The total distance-strength connection population comprises all network edges per group. For each group comparison, 10,000 bootstrapping iterations were run, linear

model coefficients were estimated and the coefficient histograms were analysed to assess if these coefficients, which represent differences in slope and intercept, were significantly different from zero (p-value < 0.05).

*Modular variability and modular dissociation.* Modularity is defined as the ability of the system to be decomposed into communities or subsystems[11]. Functional modules are highly variable between and within individuals[28], and previous investigations have proposed novel methods to study network modules, and their variability or heterogeneity in shape across time[16] and even neuroimaging modalities[71]. Here, we are proposing two approaches to study modular variability (MV) between groups and between network construction methods (local and global threshold) which we have named modular dissociation (MD).

MV measures the module affiliation variability of a respective node $s$[72]. Given a node $s$ and two modular partitions obtained from two participants or methods $i,j$, its variability $MV_s(i,j)$ is defined as:[28]

$$MV_s(i,j) = 1 - \frac{|X_s(i) \cap X_s(j)|}{|X_s(i)|} \times \frac{|X_s(i) \cap X_s(j)|}{|X_s(j)|} \qquad (3)$$

where $|X_s(i) \cap X_s(j)|$ represents the number of nodes in common between communities $X_s(i)$ and $X_s(j)$, and $|X_s(i)|$ represents the number of nodes within community $X_s(i)$, where node $s$ belongs to in the network $i$ and similarly for $|X_s(j)|$. MV ranges from 0 to 1, where 1 means no overlap between communities and 0 means no change in the node-set comprising both communities[72].

MD measures the variability between community definitions of two networks from the same participant but constructed using two thresholding regimes, i.e. modular variability between two networks constructed by local and global thresholding methods. Computation of MD is the same as for MV, but between network construction methods and consequently, MD will measure community allegiance between a global threshold which allows the strongest connections within the weighted matrix and a local threshold regime which allows weaker connections but favours community segregation by connecting the nearest neighbours. Also, note that both networks will have by definition the same edge density or cost. A toy example of MD estimation is shown in Supplementary Fig 1, and the detailed estimation procedure is shown in Fig. 1.

**Statistical analysis and reproducibility.** MV was assessed between study groups using locally thresholded networks[12]. This was decided because the modularity Q is not statistically different between groups in the NCL and NKI cohorts using this network construction method. Furthermore, the optimal edge density is reached a higher connectivity cost and led to higher Q statistics, compared with the global threshold, for the three databases in this study, see Supplementary Fig 2.

The analyses in this investigation are static analyses, which differs from previous investigations where MV has been assessed as the community change across time[10]. In order to test MV between two groups, we estimated the group's community consensus[73] with the Network Community Toolbox (NCT, http://commdetect.weebly.com/). This approach is more accurate than estimating communities from the group's mean weighted connectivity matrix. The consensus community is estimated from the optimal communities obtained with Louvain's algorithm, which was applied to all participants from the two groups. Then, MV for each participant and node is estimated as the difference between the participant's Louvain community and the two-group consensus community. However, since none of our groups is perfectly balanced, i.e. we do not have an equal number of participants in all groups, there is a high risk that the consensus community is biased towards the largest group. To resolve this, we estimated MV with a bootstrapping approach. Having two groups of size $A$ and $B$ were $B < A$, we randomly choose without replacement $B$ participants from each group, their consensus community estimated and nodal MV computed for each participant. This procedure was repeated 500 times. At each of these bootstrapped iterations, nodal MV differences between groups were assessed with nonparametric permutations (5000 permutations) after regressing out age, sex and study covariates. For the NKI and TFC cohorts, the age covariate was demeaned within groups only, in order to correct for age variances without losing the group effect (YA vs. OA). For the TFC cohort, study covariates could only be regressed within OA and within YA groups since neuroimaging acquisition for these two groups were recorded at different sites[22,57]. For the NCL cohort, we added the study covariate, and the NKI analysis did not use a covariate for different studies. All results from the 500 iterations were averaged to obtain meta-results: $p\text{-value}_{meta}$ and meta Z-score differences $Z_{meta}$. For the Z scores, the mean and variance from the nonparametric permutations were used for normalisation.

From the previous analyses, a meta-consensus community was estimated for each of the study cohorts. For this, a community consensus was estimated again from all 500 consensus communities that resulted from the bootstrapping operation.

In order to estimate between-group differences for MD, the bootstrapping approach is not necessary because MD is computed within participants. Differences between groups for MD were assessed directly with nonparametric permutations (5000 permutations) after regressing out age, sex and study covariates in a similar fashion as MV. MV and MD results were corrected for multiple testing using false discovery rate (FDR) with the Benjamini and Hochberg procedure in Matlab; mafdr.m function at p-value < 0.05 for significance level.

Additionally, mean modular comparisons were also investigated relative to the OA groups; $MD_{/OA}$. For this, nodal MD values from a group of interest were concatenated with values from the OA group (from the same cohort) to create a two-group MD vector $[G_1\ G_2]$ per node, where $G_1$ represents the OA values. Age, sex and study covariates were regressed out and a residual node vector $[R_1\ R_2]$ was obtained. From this regression, mean nodal MD was estimated and values referenced to the mean OA residuals; $[R_1\ R_2] - E(R_1)$, where $E$ stands for expected value. This latter step allows to compare MD between different cohorts, e.g. NCL and NKI, and define $MD_{/OA} = MD_{R2} - MD_{R1}$, where $MD_{R1}$ are the regressed nodal MD values of the OA groups and $MD_{R2}$ are the nodal MD values of the second group of interest. Differences between OA and the groups of interest were assessed with nonparametric two-sided rank-sum tests; here nodes were treated as independent observations and results corrected for multiple tests.

Participant motion within the MRI scanner may drive spurious results when comparing young vs older adults; it is known that older adults tend to move more than young adults during the acquisition of neuroimages. The motion was assessed by estimating Framewise displacement, FD[74] which integrates the six motion parameters estimated during the motion correction step by FSL-FLIRT. Framewise displacement is defined as:

$$FD_i = |\triangle x_i| + |\triangle y_i| + |\triangle z_i| + |\triangle a_i| + |\triangle b_i| + |\triangle c_i| \qquad (4)$$

where $\triangle x_i = x_{i-1} - x_i$ and similarly for the other rigid body parameters $[y_i, z_i, a_i, b_i, c_i]$. Rotational displacement was calculated on a surface of a 65-mm radius sphere. In order to test the motion effect, the mean FD was added as a covariate of no interest together with sex and gender in the replication analysis. Besides, a 451-ROI atlas for NKI and NCL cohorts at edge density 3.24% were also used for comparison. More details are shown in Supplementary Discussion.

The use of in-house parcellations assure there is no bias introduced by including the same cohorts under analysis (NCL and NKI) for atlas estimation and no bias introduced by the study site. Although evidence has shown that the measures of network segregation and integration seem to be robust to the underlying parcellations[75], it's unclear if the findings are also obvious in well-established brain parcellations. To this end, a replication analysis was conducted by using a multi-modal brain atlas–Human Brainnectome Atlas[52] which in total included 210 cortical, 36 subcortical and 28 cerebellar subregions. The mean FD, sex, and age were added as covariates of no interest in analysis. More details are present in Supplementary Discussion.

*Visualisation of results.* Sphere brains were plotted using the BrainNet Viewer Toolbox[76]. For both MD and MV, results were nonlinearly mapped for visualisation, and visualisation only, with an exponential function:

$$MD' = 100^{MD} \qquad (5)$$

$$MV' = 100^{MV} \qquad (6)$$

where the prime symbol indicates visualised. This transformation mapped results from a 0–1 scale to a nonlinear 1–100 scale. A similar approach was followed for the visualisation of the between-group comparison maps:

$$pvalue' = 100^{(1-pvalue)} \qquad (7)$$

Colours were implemented with diverging colour maps from blue to red colours[77].

Identification of significant structures in MNI space was implemented with the brain anatomical database within the xjView toolbox (http://www.alivelearn.net/xjview/), and the cuixufindstructure.m function in Matlab.

**Reporting summary.** Further information on research design is available in the Nature Research Reporting Summary linked to this article.

## Data availability

The Enhanced Nathan Kline Institute (NKI) Rockland Sample is available at http://fcon_1000.projects.nitrc.org/indi/enhanced/index.html, and the 1000 Functional Connectome (TFC) resting-state matrices are publicly available at http://umcd.humanconnectomeproject.org. The source data of graphs and charts are shown in Supplementary_Data.zip. Other data are available from the corresponding authors (wmtmdlove@163.com; Joseph.necus@nottingham.ac.uk) and senior author john-paul.taylor@newcastle.ac.uk on reasonable request.

## Code availability

Matlab codes for MV, MD estimation and network construction by local thresholding are available at https://github.com/LuisPerazaRo/modulardissociation and by contacting the corresponding authors wmtmdlove@163.com; Joseph.necus@nottingham.ac.uk and senior author john-paul.taylor@newcastle.ac.uk. All software used within this study is stated in the Methods sections. With exception of Matlab, all software used is free and all Matlab toolboxes are open source.

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

## Acknowledgements

The research was funded by a grant from Alzheimer's Research UK in partnership with Hidden Hearing (ARUK-PPG2016A-2) to Luis R Peraza, Marcus Kaiser, and John-Paul Taylor, and supported by the National Institute of Health Research (NIHR) Biomedical Research Centre (BRC) at Newcastle University. The study participant recruitment and data collection at Newcastle University— was funded by an intermediate clinical Wellcome Trust Fellowship (WT088441MA) to John-Paul Taylor. Marcus Kaiser and Joe Necus were supported by a Medical Research Council (MRC) grant (MR/T004347/1). Marcus Kaiser was supported by the Guangci Professorship Program of Ruijin Hospital (Shanghai Jiao Tong University). Xue Chen was supported by the China Scholarship Council (201706450045) and the Fundamental Research Funds for the Central Universities (Grant No.16CX06050A). Yanjiang Wang was supported by the China Scholarship Council (201306455001), the National Natural Science Foundation of P.R. China (Grant No.62072468) and the National Natural Science Foundation of Shandong Province (Grant No. ZR2018MF017).

## Author contributions

L.R.P. conceived the hypothesis, designed the experiments, pre-processed the neuroimaging databases, perform the research, designed data visualisation, and wrote the manuscript. X.C. performed the data analysis and wrote the manuscript. J.N., R.M. and Y.W. performed the data analysis. A.B. designed the neuroimaging acquisition protocols for the NCL databases. M.K. supervised the network connectivity analysis. J.T.O' and J.P.T performed participant recruitment, clinical assessments and diagnosis for the NCL databases. M.K. and J.P.T. oversaw the research and reviewed the manuscript. M.K. and J.P.T. shared senior authorship.

## Competing interests

The authors declare no competing interests.
