## [Peer Review File · Communications Biology]

Reviewers' comments:

Reviewer #1 (Remarks to the Author):

Thank you for inviting me to review this manuscript by Peraza and colleagues, in which the authors analyze modular variability and dissociation in networks derived from resting state data across the lifespan, and also in a range of neurodegenerative disorders. The primary novelty of the study is the introduction of a term, modular dissociation, which MD measures the distance between different means for thresholding a connectivity matrix – namely, between global and local thresholding. The authors argue that high MD values indicate that the global topological organization of the network is dissociated from local organizational structure of the network. They go on to show that MD decreases over the course of the lifespan, whereas this basic pattern is altered in neurodegenerative disease.

- In the Results section, Qrand is introduced without any explanation. I found the relevant information in the Methods section, but I suggest that some reference be made to the methodology in the Results such that the reader can properly interpret the approach.
- How many subjects were used in each bootstrap estimate? And how many iterations were used?
- In general, I found the figures to be quite visually arresting, but perhaps erring on the side of containing too much information to be truly informative.
- The authors may wish to comment on recent work that has examined functional topological structure in Parkinson's disease (e.g. Kim et al., 2018 - Brain).

Reviewer #2 (Remarks to the Author):

Peraza et al investigated the modular structure of functional connectivity graphs in the human brain, particularly under aging and neurodegenerative conditions. They used Louvain's estimations of the Brain Connectivity Toolbox to characterize modularity variability and modularity dissociation (a new metric introduced by this group) in several aging cohorts, including Alzheimer's disease (n=44), Lewy body dementia (n=42) and Parkinson disease (n=17) groups. Authors found high modular variability within the association cortices across cohorts. Moreover, they observed that aging might favors segregation and heterogeneity of brain modules. Although most of the graph theory approaches employed in this study are sound, this reviewer has some specific concerns/suggestions that may ameliorate the interpretability of the manuscript.

Major issues

As authors are probably aware of, the fMRI research community is today very worried about the introduction of micro-movements in neuroimaging data (Koene et al. and Power et al). Micro-movements induce artificial changes in functional connectivity able to modify the modular structure of graphs, particularly in aging groups compared to young groups. In its present form, the text in the manuscript does not clarify if a scrubbing protocol has been applied to the data to avoid this issue.

Anyhow, my main concern about this work relates to the use of absolute values of Pearson connectivity matrices. It is unclear why authors have converted the negative values between ROIs to positive ones. It is quite striking that authors have equalized such antagonistic relationships between areas of the brain. This analytical step may have changed the entire landscape of graph modularity. Which is the rationale to assume that a positive connection is the same as a negative one? I think authors should prove that they could obtain similar results but using alternative association matrix thresholding approaches, without converting values to absolute values.

Minor issue

At moments, the manuscript is quite dense in terminology and speculative interpretations, which makes the final message difficult to grasp. Just an example, if MD is the same as MV but when comparing global and local connectivity thresholds, why is necessary different terms for them.

Reviewer #3 (Remarks to the Author):

This study examined the brain segregated modular connectivity in two ageing cohorts and a mixed cohort of neurodegenerative diseases using BOLD-fMRI. The results revealed that the brain follows a universal pattern of modular variability, and that in ageing the brain moves towards a modular segregated structure despite presenting with increased modular heterogeneity. In the presence of neurodegeneration, the brain maintains its segregated connectivity globally but not locally; the modular brain shows patterns of differentiated pathology. The topic is interesting, but I have some concerns as follows:

- 1) Please check the NKI sample size: In the Introduction section, N=297; in the Results section, N=152+146; in the Methods section, N=151+(151-5).
- 2) How did the authors consider the signals with high frequency of > 0.1/0.08 Hz? As known, the high-frequency signals are always removed for noisy effect.
- 3) How about the differences of head motion between groups before and after despiking? Recent studies demonstrated that head motion may have both noisy and neuronal effect on functional connectivity measures (Zeng et al., PNAS, 2014; Satterthwaite et al., NeuroImage, 2012). Motion could reduce network modularity, and head motion often demonstrates systematic group effects when contrasting different populations in ageing studies (Andrews-Hanna et al., Neuron, 2007; Damoiseaux et al., Cereb Cortex, 2008).
- 4) The voxel size was 4x4x4 mm after normalization in the data preprocessing, but the authors performed spatial smoothing with 6-mm FWHM Gaussian kernel. The kernel width may be relatively small.
- 5) The authors stated that the functional brain from the independent group was parcellated with the pyClusterROI software. It means that the authors conducted all analyses using their in-house parcellation? What's the motivation of using the in-house parcellations derived from a small dataset of 29 subjects? In addition, the 29 subjects were all healthy older adults (more information of these subjects should be provided), was there any bias introduced in the comparison between young and older adult groups?
- 6) They stated that after neuroimaging pre-processing, two AD and four DLB participants from the NCL and five OA participants from the NKI cohort were excluded due to coregistration inaccuracies. How did they quantify coregistration accuracy?
- 7) I agree with the authors that previous evidence have proved the biological origin and importance of negative correlations in the brain. In the study, the authors took the absolute value of the connectivity matrices. But how did they differentiate the nodes between default mode and dorsal attention networks and arrive at a conclusion of decreased MD in the default mode module in the presence of neurodegenerative dementia, as the two networks are always strongly anti-correlated with each other (Fox et al., PNAS, 2005)?
- 8) I notice that the two curves of NKI are different from those of TFC in Figure S2 (B vs. C). In the NKI cohort, the blue curve is above the red one, but the opposite is true in the TFC cohort. Good replication between the two cohorts is most important to illuminate the reliability of the methods.

References:

- Satterthwaite TD et al. (2012) Impact of in-scanner head motion on multiple measures of functional connectivity: Relevance for studies of neurodevelopment in youth. *NeuroImage* 60:623-632.
- Zeng L-L, et al. (2014) Neurobiological basis of head motion in brain imaging. *Proc Natl Acad Sci USA*, 111(16):6058-6062.
- Damoiseaux JS et al. (2008) Reduced resting-state brain activity in the "default network" in normal aging. *Cereb Cortex* 18:1856-1864.
- Andrews-Hanna JR et al. (2007) Disruption of large-scale brain systems in advanced aging. *Neuron* 56:924-935.
- Fox MD et al. (2005) The human brain is intrinsically organized into dynamic, anticorrelated functional networks. *Proc Natl Acad Sci USA* 102:9673-9678.

Authors: We thank all reviewers for their comments and suggestions. Their comments have made our paper clearer to our readers and strengthen the results of our investigation. In the following pages, we respond to each of the reviewers' comments, and we also indicate the actions taken for each of the corrections and clarifications that were requested. Thank you all.

Reviewer #1 (Remarks to the Author):

Thank you for inviting me to review this manuscript by Peraza and colleagues, in which the authors analyze modular variability and dissociation in networks derived from resting state data across the lifespan, and also in a range of neurodegenerative disorders. The primary novelty of the study is the introduction of a term, modular dissociation, which MD measures the distance between different means for thresholding a connectivity matrix – namely, between global and local thresholding. The authors argue that high MD values indicate that the global topological organization of the network is dissociated from local organizational structure of the network. They go on to show that MD decreases over the course of the lifespan, whereas this basic pattern is altered in neurodegenerative disease.

- In the Results section, Q_{rand} is introduced without any explanation. I found the relevant information in the Methods section, but I suggest that some reference be made to the methodology in the Results such that the reader can properly interpret the approach.

Authors: We thank the reviewer for this observation regarding the definition of Q_{rand} . We have now included a brief definition of Q and Q_{rand} within the main manuscript at the first mention of these terms. We also pointed the reader to the Methods section for further reading about these two modularity statistics. Specifically, for this new submission we added the following text:

“Fig 2C shows the estimation of $Q - Q_{rand}$ (the modularity index difference between the participant’s network modularity, Q , and that of an equivalent random network Q_{rand} ,¹ see Methods section) across a range of network edge densities for the NCL cohort, and which reached its maximum at 3.24% for the 451-ROI atlas; this edge density is referred as optimal density (see methods section).” - page 6

- How many subjects were used in each bootstrap estimate? And how many iterations were used?

Authors: Thank you for these comments which allowed to clarify this point to our future readers. All subjects were used for the bootstrapping estimates in the *connection length and strength comparisons* section. However, for the analysis in this section, the connections were randomly selected at the group level; 20% of all connections within each group. Hence, the reader can quickly estimate this number for all comparisons. Each participant's network has $d \cdot M \cdot (M-1) / 2$ connections, where d is the density of the network (ranging from 0 to 1), and M is the number of nodes or brain regions in the functional atlas. As a consequence, the total population of connections is $N \cdot \text{connections}$ where N is the number of participants within the group. The total number used at each of the bootstrapping experiments was 20% of $N \cdot \text{Connections}$. We have now added additional text within our main manuscript describing the number of connections used at each iteration. Specifically, we included the following:

*“For each group comparison, a linear model to test for differences in the intercept and slope of the distance-vs-strength regression lines between the two groups was implemented. This model was defined as $\text{strength} \sim \beta_1 \cdot \text{distance} + \beta_2 \cdot \text{group} + \beta_3 \cdot \text{group} \cdot \text{distance} + \beta_4 \cdot \text{age} + \beta_5 \cdot \text{sex} + \beta_6 \cdot \text{study} + 1$, where the coefficients β_2 and β_3 account for group differences in intercept and slope respectively. However, because of the correlation values within the weighted connectivity matrices $|r_{ij}|$ did not follow a normal distribution, bootstrapped linear models were implemented instead². **At each iteration, 20% of the total distance-strength connection population was randomly sampled without replacement** together with their respective covariates for age, sex and study/site (if applicable). **The total distance-strength connection population comprises all network edges in a group.** **For each group comparison, 10000 bootstrapping iterations were run**, linear model coefficients estimated and the coefficient histograms analysed to assess if these coefficients, which represent differences in slope and intercept, were significantly different from zero ($p\text{-value} < 0.05$).”* – page 22

- In general, I found the figures to be quite visually arresting, but perhaps erring on the side of containing too much information to be truly informative.

Authors: We thank the reviewer for this comment. In our research work and submission, we aimed to be as transparent as possible to our potential readers by sharing without reserves our observations and results. It is in our interest that other research groups replicate our findings and this is the purpose for sharing data, Matlab code and the participant IDs from the NKI database that were included in our analysis. However, we agree that some of our images are not essential. This is the case, for instance, of Figure 5 in our previous submission and which showed mean group MD

values in all studied cohorts. We have moved this Figure to supplementary material and it is now shown as Supplementary Figure 3 (Fig S3).

- The authors may wish to comment on recent work that has examined functional topological structure in Parkinson's disease (e.g. Kim et al., 2018 - Brain).

Authors: We thank the reviewer for this suggestion. The paper by Kim et al. (2017, *Abnormal intrinsic brain functional network dynamics in Parkinson's disease*) used a different approach to us for the analysis of brain network variability. In essence, Kim et al. found that the dynamic network in patients with Parkinson's disease showed on average, and across time, lower global network efficiency. However, this efficiency showed higher variance, across time, in Parkinson's patients compared with healthy controls. Lower efficiency in global network connectivity is in agreement with lower global network modularity, which we also found in all our dementia groups (using global thresholding) and specifically for our Parkinson's disease dementia (PDD) group. The higher variance in global efficiency reported by Kim et al. also shows that the networks from PD patients are more unstable with a tendency toward inefficient communication.

Our results in the submitted manuscript show that the network modules in PDD patients are less variable between patients; in fact, it is not significantly different from the modular arrangement in healthy controls. However, in our research work, we did not assess variability or variance of modularity across time as in Kim et al. and our results may show a different perspective of two related phenomena. Specifically, Kim et al. found that the expression of the inferred State II network (a highly connected network where the SMN is the main module) is increased in PD. This SMN-State II network presented with increased expression and this may be related to our increased MD in same areas (SMN) in our PDD group; which indicates localised modular segregation in the SMN. To highlight these points, we have included now the following text within our main manuscript:

"PDD showed higher MD in motor-sensory and insular cortices as well as in the cerebellum; only occipital and frontal cortices showed a trend that favours segregation of modules. These differences may be related to the neuropathology of the Lewy body diseases affecting motor and cognitive brain functions. In a recent investigation by Kim et al.³, the authors reported an increase in network state transitions, which they called State I and II networks, in PD patients and which suggested unstable brain activity. This agrees with our finding of a higher MD in our group of PDD patients and which indicates a departure from segregated but structured brain modularity. However, more research on larger cohorts will be needed to bring more light into this." – page 14

Reviewer #2 (Remarks to the Author):

Peraza et al. investigated the modular structure of functional connectivity graphs in the human brain, particularly under aging and neurodegenerative conditions. They used Louvain's estimations of the Brain Connectivity Toolbox to characterize modularity variability and modularity dissociation (a new metric introduced by this group) in several aging cohorts, including Alzheimer's disease (n=44), Lewy body dementia (n=42) and Parkinson disease (n=17) groups. Authors found high modular variability within the association cortices across cohorts. Moreover, they observed that aging might favor segregation and heterogeneity of brain modules. Although most of the graph theory approaches employed in this study are sound, this reviewer has some specific concerns/suggestions that may ameliorate the interpretability of the manuscript.

Major issues

As authors are probably aware of, the fcMRI research community is today very worried about the introduction of micro-movements in neuroimaging data (Koene et al. and Power et al). Micro-movements induce artificial changes in functional connectivity able to modify the modular structure of graphs, particularly in aging groups compared to young groups. In its present form, the text in the manuscript does not clarify if a scrubbing protocol has been applied to the data to avoid this issue.

Authors: We thank the reviewer for this observation, which was not addressed in our original submission. We did not implement scrubbing as suggested by Power et al.⁴, we instead implemented wavelet denoising⁵, a similar method to scrubbing which filters out wavelets that fit high-frequency events in the time series; wavelet denoising works as a targeted high pass filter. Additionally, our resting state fMRI images were pre-processed with the recommended protocols and the steps included 1) motion correction with FSL-MCFLIRT, 2) time-slicing correction, 3) regression of the six motion parameters and mean CSF signal, 4) wavelet denoising⁵, 5) high-pass filtering with filter equivalent to 150 seconds, and 6) spatial filtering with a 6-mm FWHM filter.

Additionally, for this new submission, we have included a complete motion analysis, and we estimated the framewise displacement (FD) time-series for each participant. This analysis is now shown as part of our Supplementary Material, and we concluded that motion did not affect or influence our results. We also acknowledged that there is still motion influencing FC and much more work in this matter should be done in this regard. However, the assessment of motion correction neuroimaging pipelines is beyond the scope and aims of our investigation.

Anyway, my main concern about this work relates to the use of absolute values of Pearson connectivity matrices. It is unclear why authors have converted the negative values between ROIs to positive ones. It is quite striking that authors have equalised such antagonistic relationships between areas of the brain. This analytical step may have changed the entire landscape of graph modularity. Which is the rationale to assume that a positive connection is the same as a negative one? I think authors should prove that they could obtain similar results but using alternative association matrix thresholding approaches, without converting values to absolute values.

Authors: We thank the reviewer for this comment. In our previous publication on brain connectivity, Peraza et al. 2015⁶, we used the absolute value of the Pearson correlation matrices, and proved that findings network connectivity measures are invariant to this approach; i.e. Alzheimer's disease patients showed lower small-worldness as well as lower modularity. On the contrary, patients with dementia with Lewy bodies showed higher small-worldness and higher modularity⁶. In Peraza et al. 2015⁶ and this current manuscript, we justified the use of the absolute value because previous research proved that negative correlations are also of biological origin, and before this, it was common in the fMRI brain network research field to zero them out despite their biological importance.

The use of absolute Pearson correlation values is not different from wavelet correlation indices⁷⁻¹¹, which is probably the prevalent method in resting-state functional connectivity network studies. The reasoning for using wavelet correlation by some laboratories is because wavelet correlation coefficients are always positive, regardless of the negative or positive relationship between the two regions. Hence, the use of wavelet correlation dodges any discussion of negative relations in functional connectivity from its definition. Nevertheless, previous investigations using wavelet correlations have also reproduced network connectivity results in ageing and dementias¹⁰.

As part of our data-sharing agreement, our original raw connectivity matrices are made public. With these, readers can implement the varied approaches followed by each of the laboratories. To address this point, we have included the following text as part of our supplementary material within the section of *limitations and considerations*:

“Currently, there is no consensus related to the treatment of weights from connectivity matrices, and other research groups have proposed different approaches, all of them with

justifications. In our investigation, we decide to keep the negative Pearson correlation values because of the current research evidence indicating that these correlations are of biological origin, and we included their strength in our analyses by using the absolute value of the functional connectivity matrices. In a previous investigation from our research group⁶ we implemented this approach and proved that it reproduced perfectly previous findings that used different weight treatment approaches, e.g. Alzheimer's disease patients showed a lower small-worldness when compared with age-matched control participants⁶. The use of the absolute value for the Pearson correlation matrix is similar to the edge estimation by wavelet correlations¹², whose coefficients are always positive. Hence, we are confident that our approach is robust and reproducible” – page 19 of Supplementary Material.

Minor issue

At moments, the manuscript is quite dense in terminology and speculative interpretations, which makes the final message difficult to grasp. Just an example, if MD is the same as MV but when comparing global and local connectivity thresholds, why is necessary different terms for them.

Authors: We thank the reviewer for these comments. It is true as mentioned by the reviewer that the mathematical formulation of MD and MV is the same, however MV has been applied extensively in previous investigations to estimate modular variability across time, and this is probably the prevalent use of this statistic in the brain network research community; although there is one application of this statistic across individuals¹³. Here, we decided to rename the statistic for two reasons; 1) to emphasise that MD is not across time but between network construction methods (global and local threshold), and 2) to avoid confusion between MV which we estimated across individuals and MD which is estimated between network construction methods.

Reviewer #3 (Remarks to the Author):

This study examined the brain segregated modular connectivity in two ageing cohorts and a mixed cohort of neurodegenerative diseases using BOLD-fMRI. The results revealed that the brain follows a universal pattern of modular variability, and that in ageing the brain moves towards a modular segregated structure despite presenting with increased modular heterogeneity. In the presence of neurodegeneration, the brain maintains its segregated connectivity globally but not locally; the modular brain shows patterns of differentiated pathology. The topic is interesting, but I have some concerns as follows:

1) Please check the NKI sample size: In the Introduction section, N=297; in the Results section, N=152+146; in the Methods section, N=151+(151-5).

Authors: We thank the reviewer for this observation; the 152 was a typo, it should be 151, and it has been corrected.

2) How did the authors consider the signals with high frequency of $> 0.1/0.08$ Hz? As known, the high-frequency signals are always removed for noisy effect.

Authors: In our pre-processing protocol, we did not implement a band-pass filter as suggested by the reviewer. We instead implemented wavelet despiking⁵ which is a motion correction technique that subtracts, in the wavelet domain, events of high and low frequencies that are related to motion. After despiking, we implemented a high pass filter equivalent to 150 seconds. We are confident that our cleaning protocol suffices requirements of data quality. To make this clearer to our readers, we have included the following text within the main manuscript:

“In order to further correct for movement and other artifacts, the rs-fMRI time series were despiked with the BrainWavelet Toolbox⁵, which is a data-driven motion algorithm that subtracts high and low-frequency motion-related events.” – page 18 Methods section.

3) How about the differences of head motion between groups before and after despiking? Recent studies demonstrated that head motion may have both noisy and neuronal effect on functional connectivity measures (Zeng et al., PNAS, 2014; Satterthwaite et al., NeuroImage, 2012). Motion could reduce network modularity, and head motion often demonstrates systematic group effects

when contrasting different populations in ageing studies (Andrews-Hanna et al., *Neuron*, 2007; Damoiseaux et al., *Cereb Cortex*, 2008).

Authors: We thank the reviewer for this comment which is highly relevant and related to a comment from the first reviewer. For this new submission, we have now included a complete motion analysis comparing Framewise Displacement (FD, a motion index that integrates the six motion parameters inferred by MCFLIRT) between groups. Results from this analysis are shown as Supplementary Material, and indeed as predicted by the reviewer, our older adult (OA) group from the NKI cohort showed significantly higher FD values compared with young adults (YA), indicating higher motion within the MRI scanner in the OA group.

To check if this significant difference in FD between OA and YA affected our main results, we repeated the MV and MD comparisons but this time by binning group by motion; i.e. we separated OAs and YA in high and low motion subgroups and aimed to find differences within groups related to motion. Our analysis revealed that motion differences did not affect results from MV and MD estimations in our main analysis and this also indicates that our cleaning protocol was appropriate. However, we also acknowledged that motion is still an open issue in resting-state fMRI. The motion analysis is included now in this new submission within its section in the Supplementary Material.

4) The voxel size was 4x4x4 mm after normalization in the data preprocessing, but the authors performed spatial smoothing with 6-mm FWHM Gaussian kernel. The kernel width may be relatively small.

Authors: Thank you for this observation. We decided to leave the spatial filtering with a standard 6-mm FWHM because we did not want to broadly mix neighbouring regions of interest. The time series in our analysis were extracted as the average time-series from all functional voxels within each region of interest (ROI), depending on atlas. Each region of interest comprises several voxels. Our estimated functional atlases were made public and are available online as part of our data-sharing agreement.

5) The authors stated that the functional brain from the independent group was parcellated with the pyClusterROI software. It means that the authors conducted all analyses using their in-house parcellation? What's the motivation of using the in-house parcellations derived from a small dataset of 29 subjects? In addition, the 29 subjects were all healthy older adults (more information of these

subjects should be provided), was there any bias introduced in the comparison between young and older adult groups?

Authors: We thank the reviewer for this comment. We estimated four different functional parcellations or atlases using an independent group of humans; in this case, we used healthy older adults from an independent study in our institute that were scanned in the same MRI. The use of an independent cohort assures that there is no bias introduced by including the same cohorts under analysis (NCL and NKI) for atlas estimation, and the fact this group of participants were scanned using the same MRI device also avoids any bias by study site.

A recent investigation by Arslan and colleagues¹⁴ in resting state fMRI, has proved that network connectivity results are highly reproducible regardless of the atlas or parcellations; either functionally estimated or structurally predefined. Here, the authors proved that regardless of anatomical or functional atlases, these did not affect any of the network connectivity measures when a network analysis is carried out. Specifically, Arslan and colleagues mentioned the following:

“The results obtained with a linear SVM classifier do not favour any particular method, either anatomy, or data driven, to subdivide the brain into regions that would better reflect population differences. In fact, anatomical atlases, like AAL, which are purely based on anatomical landmarks, appear to perform as well as data-driven approaches, designed and tailored to fit the underlying rs-fMRI data. This could be attributed to the specific task at hand, since anatomical and, more specifically, cerebral volume differences have been reported between males and females that significantly influence the volume of white and gray matter (Leonard et al., 2008).” – Arslan et al. 2008.

The authors continued in the same section mentioning the following:

“As far as graph theoretical analysis is concerned, the measures of network segregation and integration, as well as the small-world topology, seem to be robust to the underlying parcellations. Despite that, all measures are highly susceptible to the granularity of the parcellation (i.e. the number of nodes within the network). These findings align with a previous study on structural connectivity and the sensitivity of network measures to the resolution of the parcellation scheme (Zalesky et al., 2010). The robustness of these network measures to the parcellation method renders them a convenient means for the analysis of population differences and explains their popularity in recent neuroscience studies on healthy and diseased subjects.”

Hence, we are confident that there was no bias introduced by the used of a functional atlas estimated from an independent group of older adults. Regarding this latter group, we have now as

well included basic demographic information from this independent group. Specifically, we mentioned the following within the section of Methods:

“Four functional brain parcellations or atlases were estimated with the pyClusterROI toolbox¹⁵: ≤ 100 , ≤ 200 , ≤ 250 , and ≤ 500 regions of interest (ROI) were requested to the algorithm. For this, an independent dataset of healthy older adults ($N = 29$ participants (16 males, 13 females) recruited within the north-east of England and who had neuroimages recorded using the same scanner as the NCL cohort; mean age 64.1 (standard deviation ± 7.92), and mean MMSE, mini-mental state examination, of 29 with a standard deviation ± 0.88), were pre-processed with the same pipeline as the NCL and NKI databases. The functional brain from the independent group was parcellated with the pyClusterROI software using a grey matter mask from the Harvard-Oxford atlas (from FSL) at 0% threshold, and which included the cerebellum. Details for the independent older adult database can be found in Schumacher et al.¹⁶ and Peraza, et al.¹⁷” – page 19

6) They stated that after neuroimaging pre-processing, two AD and four DLB participants from the NCL and five OA participants from the NKI cohort were excluded due to coregistration inaccuracies. How did they quantify coregistration accuracy?

Authors: We thank the reviewer for this observation and we apologise for not explaining this point further in our original submission. Participants were excluded because their brains were structurally altered either by ageing or disease. We noted this because when we extracted time series using the high-resolution functional atlas of 451 regions, some ROIs extracted flat time-series, i.e. zero-valued time series. These brains did not coregistered with the functional atlas extracted from an independent group. When inspecting the brains which presented with flat time series, we noticed that these brains were from older adults only; from the NKI and dementia cohorts. Further inquisition on the structural MRIs revealed structural alterations that could not be corrected by the FSL normalisation to the MNI space; these participants presented with structural alterations to their brains, probably driven by ageing or disease. From a total sample size of 439 brain scans in our study, it was not surprising to find eleven participants with structural abnormalities.

We have included the following text within the section of Methods clarifying this point.

“After neuroimaging pre-processing, two AD and four DLB participants from the NCL and five OA participants from the NKI cohort were excluded due to coregistration inaccuracies with the functional atlas; these older adult participants presented with structural alterations in their brains that could not be normalised to the standard MNI space.” – page 19

7) I agree with the authors that previous evidence has proved the biological origin and importance of negative correlations in the brain. In the study, the authors took the absolute value of the connectivity matrices. But how did they differentiate the nodes between default mode and dorsal attention networks and arrive at a conclusion of decreased MD in the default mode module in the presence of neurodegenerative dementia, as the two networks are always strongly anti-correlated with each other (Fox et al., PNAS, 2005)?

Authors: We thank the reviewer for these comments. We established the associations between modules and well-known resting state networks using the cortical distribution of the consensus modules shown in Figure 3, top of each panel in Figure 6. For instance, the fronto-parietal occipital module shown independently in Figure 6C, resembles the well-known DMN comprising precuneal, parietal and frontal cortical areas. The canonical resting state networks, such as the DMN, are well documented in the fMRI literature, and we believe it is not necessary to cover their definitions and structural topology within our manuscript.

8) I notice that the two curves of NKI are different from those of TFC in Figure S2 (B vs. C). In the NKI cohort, the blue curve is above the red one, but the opposite is true in the TFC cohort. Good replication between the two cohorts is most important to illuminate the reliability of the methods.

Authors: We thank the reviewer for this observation. In our original submission, we assessed this in the section of Limitations in the Supplementary Material. Specifically, we wrote the following observations from our results:

“Another consideration is the multi-site nature of the TFC database. The downloaded matrices came from nine different sites, and these were exclusive for older and younger adults; five sites comprised YA and four sites OA only. Because of this, we were only able to correct for within group studies but not when comparing OA vs YA. We believe that the global higher MV (Figure 5B) and Q (Figure S1) observed in OA compared with YA may be partially driven by study-site differences. However, the pattern of MV, MD and their

Although the shift between older and younger adults for the TFC cohort is intriguing, we cannot discard that the characteristics of the matrices downloaded may have caused this shift. 1) The OA and YA groups from the TFC were recorded at different sites, and 2) contrary to the other two cohorts (NKI and NCL), the TFC matrices had global signal regression, which was implemented by the original authors who made these matrices public¹⁸. Nevertheless and despite these differences, the patterns of MV and MD, as well as their contrasts between groups were similar to those observed in NKI cohort. This proves that our findings and methods are robust even when the fMRI matrices had an independent (and slightly different) preprocessing pipeline, different to the one implemented in-house for the NKI and NCL cohorts.

- 1 Sporns, O. & Betzel, R. F. Modular Brain Networks. *Annual review of psychology* **67**, 613-640, doi:10.1146/annurev-psych-122414-033634 (2016).
- 2 Mammen, E. in *When Does Bootstrap Work?* 106-117 (Springer, 1992).
- 3 Kim, J. *et al.* Abnormal intrinsic brain functional network dynamics in Parkinson's disease. *Brain : a journal of neurology* **140**, 2955-2967, doi:10.1093/brain/awx233 (2017).
- 4 Power, J. D., Barnes, K. A., Snyder, A. Z., Schlaggar, B. L. & Petersen, S. E. Spurious but systematic correlations in functional connectivity MRI networks arise from subject motion. *NeuroImage* **59**, 2142-2154, doi:10.1016/j.neuroimage.2011.10.018 (2012).
- 5 Patel, A. X. *et al.* A wavelet method for modeling and despiking motion artifacts from resting-state fMRI time series. *NeuroImage* **95**, 287-304, doi:<http://dx.doi.org/10.1016/j.neuroimage.2014.03.012> (2014).
- 6 Peraza, L. R., Taylor, J. P. & Kaiser, M. Divergent brain functional network alterations in dementia with Lewy bodies and Alzheimer's disease. *Neurobiology of aging* **36**, 2458-2467, doi:10.1016/j.neurobiolaging.2015.05.015 (2015).
- 7 Achard, S. & Bullmore, E. Efficiency and Cost of Economical Brain Functional Networks. *PLoS computational biology* **3**, e17, doi:10.1371/journal.pcbi.0030017 (2007).
- 8 Achard, S., Salvador, R., Whitcher, B., Suckling, J. & Bullmore, E. A resilient, low-frequency, small-world human brain functional network with highly connected association cortical hubs. *The Journal of neuroscience : the official journal of the Society for Neuroscience* **26**, 63-72, doi:10.1523/JNEUROSCI.3874-05.2006 (2006).
- 9 Alexander-Bloch, A. F. *et al.* Disrupted Modularity and Local Connectivity of Brain Functional Networks in Childhood-Onset Schizophrenia. *Frontiers in Systems Neuroscience* **4**, 147, doi:10.3389/fnsys.2010.00147 (2010).
- 10 Wang, J. *et al.* Disrupted functional brain connectome in individuals at risk for Alzheimer's disease. *Biological psychiatry* **73**, 472-481, doi:10.1016/j.biopsych.2012.03.026 (2013).
- 11 Bassett, D. S., Nelson, B. G., Mueller, B. A., Camchong, J. & Lim, K. O. Altered resting state complexity in schizophrenia. *NeuroImage* **59**, 2196-2207, doi:<https://doi.org/10.1016/j.neuroimage.2011.10.002> (2012).
- 12 Supekar, K., Menon, V., Rubin, D., Musen, M. & Greicius, M. D. Network Analysis of Intrinsic Functional Brain Connectivity in Alzheimer's Disease. *PLoS computational biology* **4**, e1000100, doi:10.1371/journal.pcbi.1000100 (2008).

- 13 Liao, X., Cao, M., Xia, M. & He, Y. Individual differences and time-varying features of modular brain architecture. *NeuroImage* **152**, 94-107, doi:10.1016/j.neuroimage.2017.02.066 (2017).
- 14 Arslan, S. *et al.* Human brain mapping: A systematic comparison of parcellation methods for the human cerebral cortex. *NeuroImage* **170**, 5-30, doi:10.1016/j.neuroimage.2017.04.014 (2018).
- 15 Craddock, R. C., James, G. A., Holtzheimer, P. E., Hu, X. P. & Mayberg, H. S. A whole brain fMRI atlas generated via spatially constrained spectral clustering. *Human brain mapping* **33**, 1914-1928, doi:10.1002/hbm.21333 (2012).
- 16 Schumacher, J. *et al.* Functional connectivity in dementia with Lewy bodies: A within- and between-network analysis. *Human brain mapping* **39**, 1118-1129, doi:10.1002/hbm.23901 (2018).
- 17 Peraza, L. R. *et al.* Intra- and inter-network functional alterations in Parkinson's disease with mild cognitive impairment. *Human brain mapping*, doi:10.1002/hbm.23499 (2017).
- 18 Biswal, B. B. *et al.* Toward discovery science of human brain function. *Proceedings of the National Academy of Sciences* **107**, 4734-4739, doi:10.1073/pnas.0911855107 (2010).

Reviewers' comments:

Reviewer #1 (Remarks to the Author):

The authors have adequately addressed my concerns.

Reviewer #2 (Remarks to the Author):

Authors have adequately responded to all my questions and they have modified the manuscript according to my suggestions. I do not have additional comments to the revised version of this work.

Reviewer #3 (Remarks to the Author):

Some of my concerns have been addressed, but the following issues should be further addressed: It is appreciated that the authors have done head motion analyzes in the revised manuscript. However, some questions arose in these analyzes. First, some subjects' head motion seems extremely high, as shown in Figure S9, but the authors did not exclude these data. Why? Second, the trends in Figure 2 in the main text are very similar to those in Figure S10 in the SI. Is there any relationship between the two observations? Third, the authors stated that high motion and low motion comparison for the MV values did not showed any significant differences between the subgroups (FDR corrected, p -value < 0.05). I argue that the threshold of significance level is too stringent to reduce false positive rate in the neurobiological findings of the current study. Fourth, the authors have demonstrated significant differences of head motion between some given subject groups, why did they not include head motion as a covariate in the functional connectivity analysis? Finally, in addition to noisy effect, the neuronal effect of head motion has not been mentioned at all (Yan et al., *NeuroImage*, 2013; Zeng et al., *PNAS*, 2014). The confounding effect of head motion is so important for the current study that they should mention it in the control analysis and limitation in the main text (not only in the SI). The authors have presented some reasons why they used the in-house parcellations in the study. But one of my major concerns is the reproducibility of such parcellations based on a so small dataset of 29 subjects. So I suggest that the authors should conduct a replication analysis using a well-established brain parcellation to convince the readers of the results. Regarding to the use of the absolute value of the connectivity matrices, the authors stated that the fronto-parietal occipital module shown independently in Figure 6C, resembles the well-known DMN comprising precuneal, parietal and frontal cortical areas. However, I am still confusing how they differentiate the nodes between default mode and dorsal attention networks, which strongly anti-correlates with each other. Without a clear definition, the neurobiological interpretation of the results is too speculative.

Manuscript ID: COMMSBIO-18-1113B

Original Article Title: "The functional brain favours segregated modular connectivity at old age unless targeted by neurodegeneration"

Referee expertise:

Referee #1: neuroimaging, parkinson's disease, cognition

Referee #2: cognition, imaging, brain networks

Referee #3: imaging, brain connectivity, neurodegeneration

Reviewers' comments:

Reviewer #1 (Remarks to the Author):

The authors have adequately addressed my concerns.

Reviewer #2 (Remarks to the Author):

Authors have adequately responded to all my questions and they have modified the manuscript according to my suggestions. I do not have additional comments to the revised version of this work.

Reviewer #3 (Remarks to the Author):

Some of my concerns have been addressed, but the following issues should be further addressed:

1 It is appreciated that the authors have done head motion analyzes in the revised manuscript. However, some questions arose in these analyzes. First, some subjects' head motion seems extremely high, as shown in Figure S9, but the authors did not exclude these data. Why?

Author response: We thank the reviewer for this suggestion. As shown in Figure S9, there are some participants with high head motion, especially for ADD and PDD patients. In order to ensure a sufficient number of subjects following exclusion in the later analysis, we used a less stringent FD threshold as some previous studies have done for neurological patients (Cheng et al., *Brain*, 2015; Du et al., *npj Schizophrenia*, 2021). Subjects with mean FD exceeding 0.5mm were excluded. To better diminish the motion effect on results, we thus further added the mean FD as a covariate of no interest together with sex and gender before analysis. The detailed information was shown in the 'Motion analysis' section in Supplementary Material.

References:

Cheng, W. , Rolls, E. T. , Gu, H. , Zhang, J. , & Feng, J. Autism: reduced connectivity between cortical areas involved in face expression, theory of mind, and the sense of self. *Brain* 138, 1382–1393 (2015).

Du, J. , Palaniyappan, L. , Liu, Z. , Cheng, W. , & Feng, J. The genetic determinants of language network dysconnectivity in drug-naïve early stage schizophrenia. *npj Schizophrenia* 7, 18 (2021).

2 Second, the trends in Figure 2 in the main text are very similar to those in Figure S10 in the SI. Is there any relationship between the two observations?

Author response: We thank the reviewer for this observation. Figure 2 in the main text depicted a declined trend of edge strength with edge distance. Long-range connections tend to have weak FC strengths. While Figure S10 in Supplementary Material also showed a decreased relationship but indicates that the motion-FC strength relation (FD-REFC) on the long-range connections were on average zero. Therefore, it could be speculated that the motion mostly affects close-range connections compared to the long-range connections. To

validate this, we included an additional analysis between the mean FC strength across the brain and mean FD. The related results were shown in the right panel of Figure S10 suggesting that the motion increased FC for short-distance edges but diminished or affected FC less for long-distance edges, which thus supports our conclusions.

3 Third, the authors stated that high motion and low motion comparison for the MV values did not show any significant differences between the subgroups (FDR corrected, p-value < 0.05). I argue that the threshold of significance level is too stringent to reduce false positive rate in the neurobiological findings of the current study.

Author response: We thank the reviewer for this suggestion. Due to the shown evidence of motion impact on the short-range connections in YA, OA and neurodegenerative dementia groups, we should acknowledge that the motion could influence our results. Therefore, as suggested by the reviewer before, we did a repeated analysis after motion correction by removing out subjects with large motion and adding mean FD as a covariate of no interests to test the motion effect, instead of comparing high motion and low motion cases directly. The related results were described in 'Motion analysis' section and Fig S11-S16 in Supplementary Material and 'The effects of participant motion and brain parcellation' section in Discussion. Compared to the work regardless of FD, the replication study after regressing out FD showed that the motion did not affect our conclusions and primary results despite some module changes, which confirms the previous analysis:

"Identical to the analysis without FD regression, the association cortices were found to have larger MV than primary cortices. The mean MD followed similar patterns to MV with low values within the motor-sensory and occipital modules. Compared with YA group, the OA group showed high variability and was more segregated with lower dissociation values. Although the modules and MD patterns within modules varied for NCL cohort, the three dementia groups were also shown not to differ in terms of global mean MD compared to OA group. And compared to OA and dementia groups, the YA group consistently showed higher MD which was mainly distributed in frontal related modules, temporal cortex and cerebellum."

4 Fourth, the authors have demonstrated significant differences of head motion between some given subject groups, why did they not include head motion as a covariate in the functional connectivity analysis?

Author response: Thank you for this suggestion. Following the reviewer's comment, we added the mean FD as a covariate of no interest in the functional connectivity analysis. The detailed information was shown in 'Motion analysis' section in Supplementary Material and 'The effects of participant motion and brain parcellation' section in Discussion.

5. Finally, in addition to noisy effect, the neuronal effect of head motion has not been mentioned at all (Yan et al., NeuroImage, 2013; Zeng et al., PNAS, 2014). The confounding effect of head motion is so important for the current study that they should mention it in the control analysis and limitation in the main text (not only in the SI).

Author response: We thank the reviewer for this comment which was not addressed in our submission. Recent studies demonstrated that head motion may have both artificial effect and neural effect on functional connectivity. To correct the motion artefacts, we used the group-level regression analysis of FD, that is added mean FD as a covariate, which was widely applied in previous studies. We mentioned this replication analysis about head motion in the 'Motion analysis' section in Supplementary Material and 'The effects of participant motion and brain parcellation' section in Discussion. Furthermore, we also wrote the neuronal effect of head motion as one limitation of our work in Discussion:

"This group-level regression analysis of FD was widely used to correct motion artefacts⁵¹, however, in addition to this noisy effect, the neural effect of head motion has been previously highlighted⁵⁰, which may lead to changes of neural activity and thus to be one limitation of our work."

6 The authors have presented some reasons why they used the in-house parcellations in the study. But one of my major concerns is the reproducibility of such parcellations based on a so small dataset of 29 subjects. So I suggest that the authors should conduct a replication analysis using a well-established brain parcellation to convince the readers of the results.

Author response: We thank the reviewer for this suggestion. Following the reviewer's comment, we added a replication analysis using a well-established brain parcellation—the Human Brainnetome Atlas (Fan et al., *Cerebral Cortex*, 2016) which in total included 210 cortical, 36 subcortical and 28 cerebellar subregions. All details were shown in 'Standard parcellation analysis' section in Supplementary Material and 'The effects of participant motion and brain parcellation' section in Discussion.

As shown in Fig S17-S18 in Supplementary Material, the modular definitions at the optimal edge density 4.40% compared to the edge density 3.24% based on the in-house parcellation with 451 ROIs. For example, the frontal-parietal module in 451-ROI case changed to be two separated modules for NCI cohort in the standard parcellation case, while two separate modules in 451-ROI case: cerebellum and occipital modules were integrated into one module for NCI cohort. Despite modular differences, the general MV and MD distribution of low and high values were maintained. When comparing OA and YA groups (Fig S19), the patterns of high MV and low MD for OA group were weakened and even diminished on the standard atlas. However, the features of both high MV and MD in insulo-opercular cortex, the trend of low MV and MD in parietal, cerebellum cortices were comparable. Similarly, healthy ageing moves towards a segregated modular structure from young age (Fig S21-B) and mean global MD in DLB and PDD groups did not differ significantly from the healthy ageing group. The ADD group, on the contrary, was found to have higher mean MD than OA, which was not shown in 451-ROI case.

The differences between standard-parcellation and in-house 451-ROI parcellation analysis mainly existed in the MV comparison between YA and OA (Fig S19-A), and the mean MD comparison between ADD and OA (Fig S21-B). The former differences could be caused by the bias of registration between standard parcellation and our study cohorts, which underwent different MRI scans and were acquired at different study sites. Another factor could be the resolution of Human Brainnetome Atlas including 274 ROIs compared to 451 ROIs of in-house atlas, which may weaken or decrease the significance because of the global mean across the region. The large similarity of decreased MV in OA group between the results on Human Brainnetome Atlas and in-house 247-ROI atlas (Fig S19A-B vs Fig S19C-D) supports this. The later differences, on the other hand, could be owing to the confounding effect of head motion and the standard atlas. Evidence has been shown in the following: the global mean MD_{/OA} for ADD group increased to be positive after regressing out head motion even though this was still not significant (Fig S15-B vs Fig 6-B); the use of the low-resolution Human Brainnetome Atlas promoted a further increase of the mean MD/OA for ADD group which turned to be significant. Overall, the standard parcellation was demonstrated to affect the comparison results definitely but the primary findings generally remained, especially for the MV/MD distribution patterns, the high MV/MD of OA group in the insulo-opercular cortex compared with YA, and local but not global changes of neurodegenerative dementia groups: DLB and PDD groups, which help convinced the previous studies based on 451-ROI atlas.

In the main text, we not only included the analysis of standard parcellation effect in Discussion but also made the following changes in the conclusion “modular dissociation is not affected by neurodegenerative dementia globally but at the modular level, which is particularly visible in DLB and PDD.” to give a more accurate description.

Fan, L., et al. "The Human Brainnetome Atlas: A New Brain Atlas Based on Connectional Architecture. *Cerebral Cortex* 26, 3508-3526, doi: 10.1093/cercor/bhw157 (2016).

7 Regarding to the use of the absolute value of the connectivity matrices, the authors stated that the fronto-parietal occipital module shown independently in Figure 6C, resembles the well-known DMN comprising precuneal, parietal and frontal cortical areas. However, I am still confusing how they differentiate the nodes between default mode and dorsal attention networks, which strongly anti-correlates with each other. Without a clear definition, the neurobiological interpretation of the results is too speculative. **Author respond: We thank the reviewer for this comments regarding the definition of DMN. To avoid of an excessive speculation, we changed the word 'DMN' to 'frontal-related modules' in the main text which has been highlighted in yellow in the main document.**

REVIEWERS' COMMENTS:

Reviewer #3 (Remarks to the Author):

All my concerns have been addressed.